# ACCELERATING SAFE REINFORCEMENT LEARNING WITH CONSTRAINT-MISMATCHED POLICIES

## ABSTRACT

We consider the problem of reinforcement learning when provided with (1) *a baseline control policy* and (2) *a set of constraints* that the controlled system must satisfy. The baseline policy can arise from a teacher agent, demonstration data or even a heuristic while the constraints might encode safety, fairness or other application-specific requirements. Importantly, the baseline policy may be sub-optimal for the task at hand, and is not guaranteed to satisfy the specified constraints. The key challenge therefore lies in effectively leveraging the baseline policy for faster learning, while still ensuring that the constraints are minimally violated. To reconcile these potentially competing aspects, we propose an iterative policy optimization algorithm that alternates between maximizing expected return on the task, minimizing distance to the baseline policy, and projecting the policy onto the constraint-satisfying set. We analyze the convergence of our algorithm theoretically and provide a finite-time guarantee. In our empirical experiments on five different control tasks, our algorithm consistently outperforms several state-of-the-art methods, achieving 10 times fewer constraint violations and 40% higher reward on average.[1]

## 1 INTRODUCTION

Deep reinforcement learning (RL) has achieved superior performance in several domains such as games (Mnih et al., 2013; Silver et al., 2016) and robotic control (Levine et al., 2016; Rajeswaran et al., 2017). However, in these complex applications, learning policies from scratch often requires tremendous amounts of time and computation power. To alleviate this issue, one would like to leverage a baseline policy available from a teacher or a previous task. However, the baseline policy may be sub-optimal for the new application and is not guaranteed to produce actions that satisfy given constraints on safety, fairness, or other costs. For instance, when you drive an unfamiliar vehicle, you do so cautiously to ensure safety, while at the same time you adapt your driving technique to the vehicle characteristics to improve your 'driving reward'. In effect, you (as the agent) gradually adapt a baseline policy (*i.e.,* prior driving skill) to avoid violating the constraints (*e.g.,* safety) while improving your driving reward (*e.g.,* travel time, fuel efficiency).

This problem is challenging because directly leveraging the baseline policy, as in DAGGER (Ross et al., 2011) or GAIL (Ho & Ermon, 2016), may result in policies that violate the constraints since the baseline is not guaranteed to satisfy them. To ensure constraint satisfaction, prior work either adds a hyper-parameter weighted copy of the imitation learning (IL) objective (*i.e.,* imitating the baseline policy) to the RL objective (Rajeswaran et al., 2017; Gao et al., 2018; Hester et al., 2018), or pre-trains a policy with the baseline policy and then fine-tunes it through RL (Mülling et al., 2013; Chernova & Thomaz, 2014). Both these approaches incur the cost of weight tuning to satisfy the cost constraint and do not ensure constraint satisfaction during training.

In this work, to learn from the baseline policy while satisfying constraints, we propose an iterative algorithm that performs policy updates in three stages. The first step updates the policy to maximize expected reward using trust region policy optimization (*e.g.,* TRPO (Schulman et al., 2015)). This can, however, result in a new intermediate policy that is too far from the baseline policy and one that may not satisfy the constraints. The second step performs a projection in policy space to control

---

[1]Code is available at: `https://sites.google.com/view/spacealgo`.

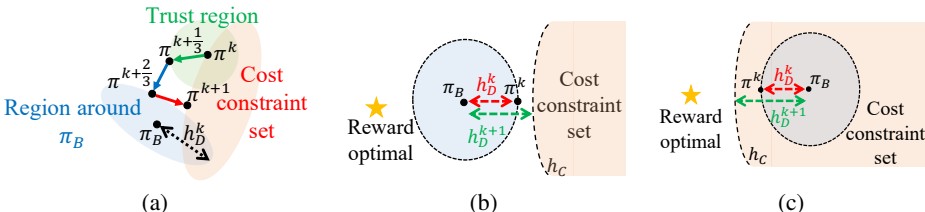

Figure 1: **(a)** Update procedures for SPACE. Step 1 (green) improves the reward in the trust region. Step 2 (blue) projects the policy onto a region around the baseline policy $\pi_B$. Step 3 (red) projects the policy onto the constraint set. **(b)** Illustrating when $\pi_B$ is *outside* the constraint set. **(c)** Illustrating when $\pi_B$ is *inside* the constraint set. The highest reward is achieved at the yellow star.

the distance between the current policy and the baseline policy. This distance is updated each episode depending on the reward improvement and constraint satisfaction, allowing the learning algorithm to explore without being overly restricted by the (potentially constraint-violating) baseline policy (Rajeswaran et al., 2017). This also enables the baseline policy to influence the learning without the computational burden of learning a cost function for the baseline policy (Kwon et al., 2020). The third step ensures constraint satisfaction at every iteration by performing a projection onto the set of policies that satisfy the given constraints. This ensures recovery from infeasible (*i.e.,* constraint-violating) states (*e.g.,* due to approximation errors), and eliminates the need for tuning weights for auxiliary cost objective functions (Tessler et al., 2018). We call our algorithm *Safe Policy Adaptation with Constrained Exploration* (SPACE).

This paper's contributions are two-fold. We first analyze our proposed SPACE algorithm and provide a finite-time guarantee for its convergence. We also provide an analysis of controlling the distance between the learned policy at iteration $k$ and the baseline policy to ensure both feasibility of the optimization problem and safe exploration by the learning agent. Second, we empirically compare SPACE with state-of-the-art algorithms on five different control tasks, including two Mujoco environments with safety constraints from Achiam et al. (2017), two challenging traffic management tasks with fairness constraints from Vinitsky et al. (2018), and one human demonstration driving task with safety constraints from Brockman et al. (2016). In all tasks, SPACE outperforms the state-of-the-art safe RL algorithm, projection-based constrained policy optimization (PCPO) in Yang et al. (2020), averaging 40% more reward with 10 times fewer cost constraint violations. This shows that SPACE leverages the baseline policy to achieve better learning efficiency while satisfying the cost constraint.

## 2 RELATED WORK

**Policy optimization with constraints.** Learning constraint-satisfying policies has been explored in the context of safe RL (Garcia & Fernandez, 2015), (Hasanbeig et al., 2020; Junges et al., 2016; Jansen et al., 2020). Prior work either uses a conditional-gradient type of approach (Achiam et al., 2017), adds a weighted copy of the cost objective in the reward function (Tessler et al., 2018; Chow et al., 2019; Fujimoto et al., 2019; Stooke et al., 2020), adds a safety layer to the policy (Dalal et al., 2018) (Avni et al., 2019), or concerns about the chanced constraints (Fu & Prashanth L, 2018; Zheng & Ratliff, 2020). Perhaps the closest work to ours is Projection-based Constrained Policy Optimization (PCPO) (Yang et al., 2020), which also uses projections in policy space to ensure constraint satisfaction. However, PCPO does not have the capability to safely exploit prior information (through a baseline policy). The lack of using prior policies in PCPO makes it sample-inefficient. In addition, our SPACE algorithm's update dynamically sets distances between policies while PCPO does not. This update is important to effectively and safely learn from the baseline policy. Furthermore, we provide a safety guarantee to ensure the feasibility of the optimization problem while PCPO does not. Merely adding an IL objective in the reward objective of PCPO cannot learn efficiently, as shown in experiments. This analysis allows us to advance towards the practical use of RL in real applications, which PCPO and other algorithms have never done before.

**Policy optimization with the initial safe set.** Wachi & Sui (2020); Sui et al. (2015); Turchetta et al. (2016) assume that the initial safe set is given, and the agent explores the environment and verifies the safety function from this initial safe set. There is no baseline policy here. In contrast, our assumption is to give a baseline policy to the agent. Both assumptions are reasonable as they provide an initial understanding of the environment.

| Algorithm | Properties | | Approach |
|---|---|---|---|
| DAGGER (Ross et al., 2011) | ⓢ | ↻ 📄📄 | Optimize IL objective |
| DAPG (Rajeswaran et al., 2017) | ⓢ | 🗄 📄📄 | Optimize RL and IL objectives jointly |
| NAC (Gao et al., 2018) | ⓢ | 🗄 📄 | Pre-train with IL and find-tune with RL |
| IC-GAIL (Wu et al., 2019) | ⓢ | 🗄 📄 | Generative adversarial IL |
| CPO (Achiam et al., 2017) | ⓢ △ | | Safe RL solved by a line search |
| RCPO (Tessler et al., 2018) | ⓢ △ | | Safe RL solved by a Lagrangian approach |
| PCPO (Yang et al., 2020) | ⓢ △ | | Safe RL solved by a projection |
| PID-RL (Stooke et al., 2020) | ⓢ △ | | Safe RL solved by PID control |
| **SPACE (ours)** | ⓢ △ | ↻ 📄 | Safe RL aided by baseline policies with analysis |

ⓢ maximize reward, △ constraint-cost satisfaction, ↻ on-policy imitation, 🗄 off-policy imitation, 📄📄 w/ expert/optimal demonstrations, 📄 w/ suboptimal demonstration

Table 1: Comparison to prior work.

**Leveraging baseline policies for RL.** Prior work has used baseline policies to provide initial information to RL algorithms to reduce or avoid undesirable situations. This is done by either: initializing the policy with the baseline policy (Driessens & Džeroski, 2004; Smart & Kaelbling, 2000; Koppejan & Whiteson, 2011; Abbeel et al., 2010; Gao et al., 2018; Le et al., 2019; Vecerik et al., 2017; Jaques et al., 2019); or providing a teacher's advice to the agent (Garcia & Fernández, 2012; Quintía Vidal et al., 2013; Abel et al., 2017; Zhang et al., 2019). However, such works often assume that the baseline policy is constraint-satisfying (Sun et al., 2018; Balakrishna et al., 2019). In contrast, our SPACE algorithm safely leverages the baseline policy without requiring it to satisfy the specified constraints.

Pathak et al. (2015); Bartocci et al. (2011) also modify the existing known models (policies) based on new conditions in the context of the formal methods. In contrast, we solve this problem using projections in the policy space.

**Learning from logged demonstration data.** To effectively learn from demonstration data given by the baseline policy, Wu et al. (2019); Brown et al. (2019); Kwon et al. (2020) assess the demonstration data by either: predicting their cost in the new task using generative adversarial networks (GANs) (Goodfellow et al., 2014); or directly learning the cost function of the demonstration data. However, these approaches require a large number of training samples from the new task. In addition, the learned cost function is not guaranteed to recover the true one. This may result in driving the agent to undesirable situations. In contrast, SPACE controls the distance between the learned and baseline policies to ensure reward improvement and constraint satisfaction.

Table 1 catalogs related work on safe RL and IL. We can differentiate these methods on several key axes: **(1)** whether they optimize reward and/or ensure cost satisfaction, **(2)** whether they use on-policy methods (*i.e.,* query the baseline policy) or off-policy methods (*i.e.,* learn from a batch of demonstration data), **(3)** whether the baseline policy or demonstration data is optimal for the agent's own objective, and **(4)** the technical approach of the work.

## 3 PROBLEM FORMULATION

We frame our problem as a constrained Markov Decision Process (CMDP) (Altman, 1999), defined as a tuple $< \mathcal{S}, \mathcal{A}, T, R, C >$. Here $\mathcal{S}$ is the set of states, $\mathcal{A}$ is the set of actions, and $T$ specifies the conditional probability $T(s'|s, a)$ that the next state is $s'$ given the current state $s$ and action $a$. In addition, $R : \mathcal{S} \times \mathcal{A} \to \mathbb{R}$ is a reward function, and $C : \mathcal{S} \times \mathcal{A} \to \mathbb{R}$ is a constraint cost function. The reward function encodes the benefit of using action $a$ in state $s$, while the cost function encodes the corresponding constraint violation penalty.

A policy is a map from states to probability distributions on $\mathcal{A}$. It specifies that in state $s$ the selected action is drawn from the distribution $\pi(s)$. The state then transits from $s$ to $s'$ according to the state transition distribution $T(s'|s, a)$. In doing so, a reward $R(s, a)$ is received and a constraint cost $C(s, a)$ is incurred, as outlined above.

Let $\gamma \in (0, 1)$ denote a discount factor, and $\tau$ denote the trajectory $\tau = (s_0, a_0, s_1, \cdots)$ induced by a policy $\pi$. Normally, we seek a policy $\pi$ that maximizes a cumulative discounted reward

$$J_R(\pi) \doteq \mathbb{E}_{\tau \sim \pi} \left[ \sum_{t=0}^{\infty} \gamma^t R(s_t, a_t) \right], \tag{1}$$

while keeping the cumulative discounted cost below $h_C$

$$J_C(\pi) \doteq \mathbb{E}_{\tau \sim \pi} \left[ \sum_{t=0}^{\infty} \gamma^t C(s_t, a_t) \right] \leq h_C. \tag{2}$$

Here we consider an additional objective. We are provided with a baseline policy $\pi_B$ and at each state $s$ we measure the divergence between $\pi(s)$ and $\pi_B(s)$. For example, this could be the KL-divergence $D(s) \doteq D_{\text{KL}}(\pi(s) \| \pi_B(s))$. We then seek a policy that maximizes Eq. (1), satisfies Eq. (2), and ensures the discounted divergence between the learned and baseline policies is below $h_D$:

$$J_D(\pi) \doteq \mathbb{E}_{\tau \sim \pi} \left[ \sum_{t=0}^{\infty} \gamma^t D(s_t) \right] \leq h_D. \tag{3}$$

We do not assume that the baseline policy satisfies the cost constraint. Hence we allow $h_D$ to be adjusted during the learning of $\pi$ to allow for reward improvement and constraint satisfaction.

Let $\mu_t(\cdot | \pi)$ denote the state distribution at time $t$ under policy $\pi$. The discounted state distribution induced by $\pi$ is defined to be $d^\pi(s) \doteq (1 - \gamma) \sum_{t=0}^{\infty} \gamma^t \mu_t(s | \pi)$. Now bring in the reward advantage function (Kakade & Langford, 2002) defined by

$$A_R^\pi(s, a) \doteq Q_R^\pi(s, a) - V_R^\pi(s),$$

where $V_R^\pi(s) \doteq \mathbb{E}_{\tau \sim \pi} \left[ \sum_{t=0}^{\infty} \gamma^t R(s_t, a_t) | s_0 = s \right]$ is the expected reward from state $s$ under policy $\pi$, and $Q_R^\pi(s, a) \doteq \mathbb{E}_{\tau \sim \pi} \left[ \sum_{t=0}^{\infty} \gamma^t R(s_t, a_t) | s_0 = s, a_0 = a \right]$ is the expected reward from state $s$ and initial action $a$, and thereafter following policy $\pi$. These definitions allow us to express the reward performance of one policy $\pi'$ in terms of another $\pi$:

$$J_R(\pi') - J_R(\pi) = \frac{1}{1-\gamma} \mathbb{E}_{s \sim d^{\pi'}, a \sim \pi'} [A_R^\pi(s, a)].$$

Similarly, we can define $A_D^\pi(s, a)$, $Q_D^\pi(s, a)$ and $V_D^\pi(s)$ for the divergence cost, and $A_C^\pi(s, a)$, $Q_C^\pi(s, a)$ and $V_C^\pi(s)$ for the constraint cost.

## 4    Safe Policy Adaptation with Constrained Exploration (SPACE)

We now describe the proposed iterative three step algorithm illustrated in Fig. 1. In what follows, $\pi^k$ denotes the learned policy after iteration $k$, and $M$ denotes a distance measure between policies. For example, $M$ might be the 2-norm of the difference of policy parameters or some average over the states of the KL-divergence of the action policy distributions.

**Step 1.** We perform one step of trust region policy optimization (Schulman et al., 2015). This maximizes the reward advantage function $A_R^\pi(s, a)$ over a KL-divergence neighborhood of $\pi^k$:

$$\pi^{k+\frac{1}{3}} = \arg \max_{\pi} \mathbb{E}_{s \sim d^{\pi^k}, a \sim \pi} [A_R^{\pi^k}(s, a)] \quad \text{s.t.} \; \mathbb{E}_{s \sim d^{\pi^k}} \left[ D_{\text{KL}}(\pi(s) \| \pi^k(s)) \right] \leq \delta. \tag{4}$$

**Step 2.** We project $\pi^{k+\frac{1}{3}}$ onto a region around $\pi_B$ controlled by $h_D^k$ to minimize $M$:

$$\pi^{k+\frac{2}{3}} = \arg \min_{\pi} \; M(\pi, \pi^{k+\frac{1}{3}}) \quad \text{s.t.} \; J_D(\pi^k) + \frac{1}{1-\gamma} \mathbb{E}_{s \sim d^{\pi^k}, a \sim \pi} [A_D^{\pi^k}(s)] \leq h_D^k. \tag{5}$$

**Step 3.** We project $\pi^{k+\frac{2}{3}}$ onto the set of policies satisfying the cost constraint to minimize $M$:

$$\pi^{k+1} = \arg \min_{\pi} \; M(\pi, \pi^{k+\frac{2}{3}}) \quad \text{s.t.} \; J_C(\pi^k) + \frac{1}{1-\gamma} \mathbb{E}_{s \sim d^{\pi^k}, a \sim \pi} [A_C^{\pi^k}(s, a)] \leq h_C. \tag{6}$$

**Control $h_D^k$ in Step 2.** We select $h_D^0$ to be small and gradually increase $h_D^k$ at each iteration to expand the region around $\pi_B$. Specifically, we make $h_D^{k+1} > h_D^k$ if:

(a) $J_C(\pi^k) > J_C(\pi^{k-1})$: this increase is to ensure a nonempty intersection between the region around $\pi_B$ and the cost constraint set (feasibility). See Fig. 1(b).

(b) $J_R(\pi^k) < J_R(\pi^{k-1})$: this increase gives the next policy more freedom to improve the reward and the cost constraint performance (exploration). See Fig. 1(c).

It remains to determine how to set the new value of $h_D^{k+1}$. Let $\mathcal{U}_1$ denote the set of policies satisfying the cost constraint, and $\mathcal{U}_2^k$ denote the set of policies in the region around $\pi_B$ controlled by $h_D^k$. Then we have the following Lemma.

**Lemma 4.1** (**Updating** $h_D$). *If at step* $k+1$: $h_D^{k+1} \geq \mathcal{O}\big((J_C(\pi^k)-h_C)^2\big)+h_D^k$, *then* $\mathcal{U}_1 \cap \mathcal{U}_2^{k+1} \neq \emptyset$ *(feasibility) and* $\mathcal{U}_2^{k+1} \cap \partial \mathcal{U}_1 \neq \emptyset$ *(exploration).*

*Proof.* See the supplementary material. $\square$

Lemma 4.1 ensures that the boundaries of the region around $\pi_B$ determined by $h_D$ and the set of policies satisfying the cost constraint intersect. Note that $h_D$ will become large enough to guarantee feasibility during training. This allows the learning algorithm to explore policies within the cost constraint set while still learning from the baseline policy.

## 5 A THEORETICAL ANALYSIS OF SPACE

We will implement a policy as a neural network with fixed architecture parameterized by $\boldsymbol{\theta} \in \mathbb{R}^n$. We then learn a policy from the achievable set $\{\pi(\cdot|\boldsymbol{\theta}): \boldsymbol{\theta} \in \mathbb{R}^n\}$ by iteratively learning $\boldsymbol{\theta}$. Let $\boldsymbol{\theta}^k$ and $\pi^k \doteq \pi(\cdot|\boldsymbol{\theta}^k)$ denote the parameter value and the corresponding policy at step $k$. In this setting, it is impractical to solve for the policy updates in Eq. (4), (5) and (6). Hence we approximate the reward function and constraints with first order Taylor expansions, and KL-divergence with a second order Taylor expansion. We will need the following derivatives: **(1)** $\boldsymbol{g}^k \doteq \nabla_{\boldsymbol{\theta}} \mathbb{E}_{s \sim d^{\pi^k}, \, a \sim \pi}[A_R^{\pi^k}(s,a)]$, **(2)** $\boldsymbol{a}^k \doteq \nabla_{\boldsymbol{\theta}} \mathbb{E}_{s \sim d^{\pi^k}, \, a \sim \pi}[A_D^{\pi^k}(s)]$, **(3)** $\boldsymbol{c}^k \doteq \nabla_{\boldsymbol{\theta}} \mathbb{E}_{s \sim d^{\pi^k}, \, a \sim \pi}[A_C^{\pi^k}(s,a)]$, and **(4)** $\boldsymbol{F}^k \doteq \nabla_{\boldsymbol{\theta}}^2 \mathbb{E}_{s \sim d^{\pi^k}} \big[D_{\mathrm{KL}}(\pi(s) \| \pi^k(s))\big]$. Each of these derivatives are taken w.r.t. the neural network parameter and evaluated at $\boldsymbol{\theta}^k$. We also define $b^k \doteq J_D(\pi^k) - h_D^k$, and $d^k \doteq J_C(\pi^k) - h_C$. Let $u^k \doteq \sqrt{\frac{2\delta}{\boldsymbol{g}^{k^T} \boldsymbol{F}^{k-1} \boldsymbol{g}^k}}$, and $\boldsymbol{L} = \boldsymbol{I}$ for the 2-norm projection and $\boldsymbol{L} = \boldsymbol{F}^k$ for the KL-divergence projection.

We first approximate Eq. (4), Eq. (5) and Eq. (6) using the following approximations:

**Step 1.** Approximating Eq. (4) yields

$$\boldsymbol{\theta}^{k+\frac{1}{3}} = \arg\max_{\boldsymbol{\theta}} \boldsymbol{g}^{k^T}(\boldsymbol{\theta} - \boldsymbol{\theta}^k) \quad \text{s.t. } \frac{1}{2}(\boldsymbol{\theta} - \boldsymbol{\theta}^k)^T \boldsymbol{F}^k (\boldsymbol{\theta} - \boldsymbol{\theta}^k) \leq \delta. \tag{7}$$

**Step 2 and Step 3.** Approximating Eq. (5) and (6), similarly yields

$$\boldsymbol{\theta}^{k+\frac{2}{3}} = \arg\min_{\boldsymbol{\theta}} \frac{1}{2}(\boldsymbol{\theta} - \boldsymbol{\theta}^{k+\frac{1}{3}})^T \boldsymbol{L}(\boldsymbol{\theta} - \boldsymbol{\theta}^{k+\frac{1}{3}}) \quad \text{s.t. } \boldsymbol{a}^{k^T}(\boldsymbol{\theta} - \boldsymbol{\theta}^k) + b^k \leq 0, \tag{8}$$

$$\boldsymbol{\theta}^{k+1} = \arg\min_{\boldsymbol{\theta}} \frac{1}{2}(\boldsymbol{\theta} - \boldsymbol{\theta}^{k+\frac{2}{3}})^T \boldsymbol{L}(\boldsymbol{\theta} - \boldsymbol{\theta}^{k+\frac{2}{3}}) \quad \text{s.t. } \boldsymbol{c}^{k^T}(\boldsymbol{\theta} - \boldsymbol{\theta}^k) + d^k \leq 0, \tag{9}$$

where $\boldsymbol{L} = \boldsymbol{I}$ for the 2-norm projection and $\boldsymbol{L} = \boldsymbol{F}^k$ for the KL-divergence projection. We solve these problems using using convex programming (see the supplementary material for the derivation), then for each policy update, we have

$$\begin{aligned}
\boldsymbol{\theta}^{k+1} = \boldsymbol{\theta}^k + u^k \boldsymbol{F}^{k-1} \boldsymbol{g}^k - \max(0, \frac{u^k \boldsymbol{a}^{k^T} \boldsymbol{F}^{k-1} \boldsymbol{g}^k + b^k}{\boldsymbol{a}^{k^T} \boldsymbol{L}^{-1} \boldsymbol{a}^k}) \boldsymbol{L}^{-1} \boldsymbol{a}^k \\
- \max(0, \frac{u^k \boldsymbol{c}^{k^T} \boldsymbol{F}^{k-1} \boldsymbol{g}^k + d^k}{\boldsymbol{c}^{k^T} \boldsymbol{L}^{-1} \boldsymbol{c}^k}) \boldsymbol{L}^{-1} \boldsymbol{c}^k.
\end{aligned} \tag{10}$$

Algorithm 1 shows the corresponding pseudocode.

**Convergence analysis.** We consider the following simplified problem to provide a finite-time guarantee of SPACE:

$$\min_{\boldsymbol{\theta} \in \mathcal{C}_1 \cap \mathcal{C}_2} f(\boldsymbol{\theta}), \tag{11}$$

where $f : \mathbb{R}^n \to \mathbb{R}$ is a twice continuously differentiable function at every point in a open set $\mathcal{X} \subseteq \mathbb{R}^n$, and $\mathcal{C}_1 \subseteq \mathcal{X}$ and $\mathcal{C}_2 \subseteq \mathcal{X}$ are compact convex sets with $\mathcal{C}_1 \cap \mathcal{C}_2 \neq \emptyset$. The function $f$ is the

negative reward function of our CMDP, and the two constraint sets represent the cost constraint set and the region around $\pi_B$.

For a vector $\boldsymbol{x}$, let $\|\boldsymbol{x}\|$ denote the Euclidean norm. For a matrix $\boldsymbol{M}$ let $\|\boldsymbol{M}\|$ denote the induced matrix 2-norm, and $\sigma_i(\boldsymbol{M})$ denote the $i$-th largest singular value of $\boldsymbol{M}$.

**Assumption 1.** We assume:

(1.1) The gradient $\nabla f$ is $L$-Lipschitz continuous over a open set $\mathcal{X}$.

(1.2) For some constant $G$, $\|\nabla f(\boldsymbol{\theta})\| \leq G$.

(1.3) For some constant $H$, $\text{diam}(\mathcal{C}_1) \leq H$ and $\text{diam}(\mathcal{C}_2) \leq H$.

Assumptions (1.1) and (1.2) ensure that the gradient can not change too rapidly and the norm of the gradient can not be too large. Assumption (1.3) implies that for every iteration, the diameter of the region around $\pi_B$ is bounded above by $H$.

We will need a concept of an $\epsilon$-first order stationary point (Mokhtari et al., 2018). For $\epsilon > 0$, we say that $\boldsymbol{\theta}^* \in \mathcal{C}_1 \cap \mathcal{C}_2$ an $\epsilon$-first order stationary point ($\epsilon$-FOSP) of Problem (11) under KL-divergence projection if

$$\nabla f(\boldsymbol{\theta}^*)^T (\boldsymbol{\theta} - \boldsymbol{\theta}^*) \geq -\epsilon, \quad \forall \boldsymbol{\theta} \in \mathcal{C}_1 \cap \mathcal{C}_2. \tag{12}$$

Similarly, under the 2-norm projection, $\boldsymbol{\theta}^* \in \mathcal{C}_1 \cap \mathcal{C}_2$ an $\epsilon$-FOSP of (11) if

$$\nabla f(\boldsymbol{\theta}^*)^T \boldsymbol{F}^* (\boldsymbol{\theta} - \boldsymbol{\theta}^*) \geq -\epsilon, \quad \forall \boldsymbol{\theta} \in \mathcal{C}_1 \cap \mathcal{C}_2, \tag{13}$$

where $\boldsymbol{F}^* \doteq \nabla_{\boldsymbol{\theta}}^2 \mathbb{E}_{s \sim d^{\pi^*}} [D_{\text{KL}}(\pi(s)\|\pi^*(s))]$. Notice that SPACE converges to distinct stationary points under the two possible projections (see the supplementary material). With these assumptions, we have the following Theorem.

**Theorem 5.1 (Finite-Time Guarantee of SPACE).** *Under the KL-divergence projection, there exists a sequence $\{\eta^k\}$ such that SPACE converges to an $\epsilon$-FOSP in at most $\mathcal{O}(\epsilon^{-2})$ iterations. Moreover, at step $k + 1$*

$$f(\boldsymbol{\theta}^{k+1}) \leq f(\boldsymbol{\theta}^k) - \frac{L\epsilon^2}{2(G + \frac{H\sigma_1(\boldsymbol{F}^k)}{\eta^k})^2}. \tag{14}$$

*Similarly, under the 2-norm projection, there exists a sequence $\{\eta^k\}$ such that SPACE converges to an $\epsilon$-FOSP in at most $\mathcal{O}(\epsilon^{-2})$ iterations. Moreover, at step $k + 1$*

$$f(\boldsymbol{\theta}^{k+1}) \leq f(\boldsymbol{\theta}^k) - \frac{L\epsilon^2}{2(G\sigma_1(\boldsymbol{F}^{k^{-1}}) + \frac{H}{\eta^k})^2}. \tag{15}$$

*Proof.* The proof and the sequence $\{\eta^k\}$ are given in the supplementary material. $\square$

We now make several observations for Theorem 5.1.

**(1)** The smaller $H$ is, the greater the decrease in the objective. This observation supports the idea of starting with a small value for $h_D$ and increasing it only when needed.

**(2)** Under the KL-divergence projection, the effect of $\sigma_1(\boldsymbol{F}^k)$ is negligible. This is because in this case $\eta^k$

---

**Algorithm 1** SPACE

> Initialize a policy $\pi^0 = \pi(\cdot|\boldsymbol{\theta}^0)$ and a trajectory buffer $\mathcal{B}$
> **for** $k = 0, 1, 2, \cdots$ **do**
>     Run $\pi^k = \pi(\cdot|\boldsymbol{\theta}^k)$ and store trajectories in $\mathcal{B}$
>     Compute $\boldsymbol{g}, \boldsymbol{a}, \boldsymbol{c}, \boldsymbol{F}, b$ and $d$ using $\mathcal{B}$
>     Obtain $\boldsymbol{\theta}^{k+1}$ using the update in Eq. (10)
>     **if** $J_C(\pi^k) > J_C(\pi^{k-1})$ or $J_R(\pi^k) < J_R(\pi^{k-1})$ **then**
>         Update $h_D^{k+1}$ using **Lemma 4.1**
>     Empty $\mathcal{B}$

---

is determined by the KL-divergence between two consecutive updated policies (see the supplementary material). This implies that $\eta^k$ is proportional to $\sigma_1(\boldsymbol{F}^k)$. Hence $\sigma_1(\boldsymbol{F}^k)$ does not play a major role in decreasing the objective value.

**(3)** Under the 2-norm projection, the smaller $\sigma_1(\boldsymbol{F}^{k^{-1}})$ (*i.e.,* larger $\sigma_n(\boldsymbol{F}^k)$) is, the greater the decrease in the objective. This is because a large $\sigma_n(\boldsymbol{F}^k)$ means a large curvature of $f$ in all directions. This implies that the 2-norm distance between the pre-projection and post-projection points is small, leading to a small deviation from the reward improvement direction after doing projections (see the supplementary material for a visualization).

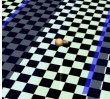 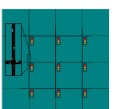 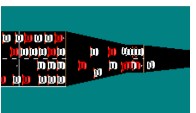 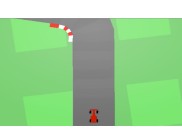 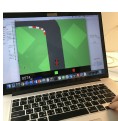

(a) Gather    (b) Circle    (c) Grid    (d) Bottleneck    (e) Car-racing    (f) Demo.

Figure 2: **(a)** Gather task: the agent is rewarded for gathering green apples, but is constrained to collect a limited number of red apples (Achiam et al., 2017). **(b)** Circle task: the agent is rewarded for moving in a specified wide circle, but is constrained to stay within a safe region smaller than the radius of the circle (Achiam et al., 2017). **(c)** Grid task: the agent controls the traffic lights in a grid road network and is rewarded for high throughput, but is constrained to let lights stay red for at most 7 consecutive seconds (Vinitsky et al., 2018). **(d)** Bottleneck task: the agent controls a set of autonomous vehicles (shown in red) in a traffic merge situation and is rewarded for achieving high throughput, but constrained to ensure that human-driven vehicles (shown in white) have low speed for no more than 10 seconds (Vinitsky et al., 2018). **(e)** Car-racing task: the agent controls an autonomous vehicle on a race track and is rewarded for driving through as many tiles as possible, but is constrained to use the brakes at most 5 times to encourage a smooth ride (Brockman et al., 2016). **(f)** A human player plays car-racing with demonstration data logged.

## 6 EXPERIMENTS

**Tasks.** We compare the proposed algorithm with existing approaches on five control tasks: three tasks with safety constraints ((a), (b) and (e) in Fig. 2), and two tasks with fairness constraints ((c) and (d) in Fig. 2). These tasks are briefly described in the caption of Fig. 2. We chose the traffic management tasks since a good control policy can benefit millions of drivers. In addition, we chose the car-racing task since a good algorithm should safely learn from baseline human policies. For all the algorithms, we use neural networks to represent Gaussian policies. We use the KL-divergence projection in the Mujoco and car-racing tasks, and the 2-norm projection in the traffic management task since it achieves better performance. We use a grid-search to select for the hyper-parameters. See Appendix for more experimental details.

**Baseline algorithms.** We compare SPACE with five baseline methods.
**(1)** *Fixed-point Constrained Policy Optimization (f-CPO)*. In f-CPO, the weight $\lambda$ is fixed followed by a CPO update (Achiam et al., 2017). The f-CPO policy update solves:

$$\boldsymbol{\theta}^{k+1} = \arg\max_{\boldsymbol{\theta}}(\boldsymbol{g}^k + \lambda \boldsymbol{a}^k)^T(\boldsymbol{\theta} - \boldsymbol{\theta}^k) \text{ s.t.} \tfrac{1}{2}(\boldsymbol{\theta} - \boldsymbol{\theta}^k)^T \boldsymbol{F}^k(\boldsymbol{\theta} - \boldsymbol{\theta}^k) \leq \delta, \ \boldsymbol{c}^{k^T}(\boldsymbol{\theta} - \boldsymbol{\theta}^k) + d^k \leq 0.$$

f-CPO adds the divergence objective in the reward function. The lack of weight-adjustment makes it susceptible to suboptimal $\pi_B$.
**(2)** *Fixed-point PCPO (f-PCPO)*. In f-PCPO, the weight $\lambda$ is fixed followed by a PCPO update (Yang et al., 2020). The f-PCPO policy update solves:

$$\boldsymbol{\theta}^{k+\frac{1}{2}} = \arg\max_{\boldsymbol{\theta}} \ (\boldsymbol{g}^k + \lambda \boldsymbol{a}^k)^T(\boldsymbol{\theta} - \boldsymbol{\theta}^k) \quad \text{s.t. } \tfrac{1}{2}(\boldsymbol{\theta} - \boldsymbol{\theta}^k)^T \boldsymbol{F}^k(\boldsymbol{\theta} - \boldsymbol{\theta}^k) \leq \delta,$$

$$\boldsymbol{\theta}^{k+1} = \arg\min_{\boldsymbol{\theta}} \ \tfrac{1}{2}(\boldsymbol{\theta} - \boldsymbol{\theta}^{k+\frac{1}{2}})^T \boldsymbol{L}(\boldsymbol{\theta} - \boldsymbol{\theta}^{k+\frac{1}{2}}) \quad \text{s.t. } \boldsymbol{c}^{k^T}(\boldsymbol{\theta} - \boldsymbol{\theta}^k) + d^k \leq 0.$$

f-PCPO also adds the divergence objective in the reward function, but followed by a PCPO update
**(3)** *Dynamic-point Constrained Policy Optimization (d-CPO)*. The d-CPO update solves f-CPO problem with a stateful $\lambda^{k+1} = (\lambda^k)^\beta$, where $0 < \beta < 1$. This is inspired by Rajeswaran et al. (2017), in which they have the same weight-scheduling method to adjust $\lambda^k$.
**(4)** *Dynamic-point PCPO (d-PCPO)*. The d-PCPO update solves f-PCPO problem with a stateful $\lambda^{k+1} = (\lambda^k)^\beta$, where $0 < \beta < 1$. d-PCPO also uses the same method to learn from $\pi_B$ as in d-CPO. For all the experiments and the algorithms, the weight is fixed and it is set to 1.

**Baseline policies** $\pi_B$**.** To test whether SPACE can safely and efficiently leverage the baseline policy, we used *three variants* of the baseline policies: **(1)** *suboptimal* $\pi_B^{\text{cost}}$ with $J_C(\pi_B^{\text{cost}}) \approx 0$, **(2)** *suboptimal* $\pi_B^{\text{reward}}$ with $J_C(\pi_B^{\text{reward}}) > h_C$, and **(3)** $\pi_B^{\text{near}}$ with $J_C(\pi_B^{\text{near}}) \approx h_C$ (*i.e.*, the baseline policy has the same cost constraint as the agent). Note that these $\pi_B$ have *different* degrees of constraint satisfaction. This is to examine **(1)** whether SPACE can achieve better learning efficiency given the constraint-satisfying $\pi_B$ (*i.e.*, $\pi_B^{\text{cost}}$ and $\pi_B^{\text{near}}$), and **(2)** whether SPACE can safely learn from constraint-violating $\pi_B$ (*i.e.*, $\pi_B^{\text{reward}}$). In addition, in the car-racing task we pre-train a $\pi_B$ using an off-policy algorithm (DDPG (Lillicrap et al., 2016)), which directly learns from *human demonstration data* (Fig. 2(f)). This suboptimal human baseline policy is denoted by $\pi_B^{\text{human}}$. This is

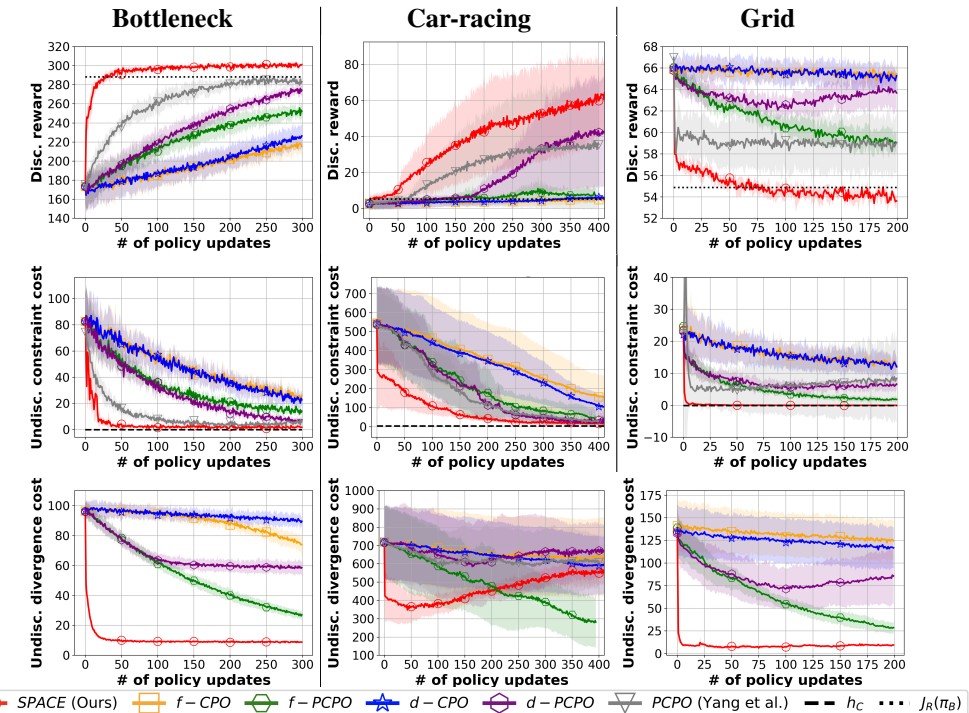

Figure 3: The discounted reward, and the undiscounted constraint cost, the undiscounted divergence cost over policy updates for the tested algorithms and tasks. The solid line is the mean and the shaded area is the standard deviation over 5 runs. The baseline policies in the grid and bottleneck tasks are $\pi_B^{\text{near}}$, and the baseline policy in the car-racing task is $\pi_B^{\text{human}}$. The black dashed line is the cost constraint threshold $h_C$. Overall, we observe that SPACE is the only algorithm that satisfies the constraints while achieving superior reward performance in all cases. Although $\pi_B^{\text{human}}$ has substantially low reward, SPACE still can learn to improve the reward. (We only show the results in these tasks as representative cases since these tasks are more challenging. Please read Appendix for more results. Best viewed in color.)

to simulate that $\pi_B$ comes from a heuristic or human demonstration data, which are far from optimal. The detailed implementation of updating $h_D$ can be found in Appendix E.1.

**Overall performance.** The learning curves of the discounted reward ($J_R(\pi)$), the undiscounted constraint cost ($J_C(\pi)$), and the undiscounted divergence cost ($J_D(\pi)$) over policy updates are shown for all tested algorithms and tasks in Fig. 3. We use $\pi_B^{\text{near}}$ in bottleneck and grid tasks, and $\pi_B^{\text{human}}$ in car-racing task. Note that $\pi_B^{\text{human}}$ is *highly suboptimal* to the agent (*i.e.*, $J_R(\pi_B^{\text{human}})$ is small). The value of the reward is only around 5 as shown in the plot. It does not solve the task at hand. Overall, we observe that **(1)** SPACE achieves at least 2 times faster cost constraint satisfaction in all cases even learning from $\pi_B^{\text{human}}$. **(2)** SPACE achieves at least 10% more reward in the bottleneck and car-racing tasks compared to the best baseline, and **(3)** SPACE is the only algorithm that satisfies the cost constraints in all cases. In contrast, PCPO agent cannot robustly satisfy the cost constraints. This is because using the baseline policy can provide additional information about the safety constraints. For example, when using $\pi_B^{\text{cost}}$ that is only optimized for the safety constraints specified by the environment, we can observe the agent trained with SPACE satisfies the constraints more. This shows that using the baseline policy could also help to improve the cost performance.

For example, in the car-racing task we observe that $J_D(\pi)$ in SPACE decreases at the initial iteration, but increases in the end. This implies that the learned policy is guided by the baseline policy $\pi_B^{\text{human}}$ in the beginning, but use less supervision in the end. In addition, in the grid task we observe that the final reward of SPACE is lower than the baseline algorithm. This is because that SPACE converges to a policy in the cost constraint set, whereas the baseline algorithms do not find constraint-satisfying policies. These observations show that SPACE can robustly ensure constraint satisfaction while achieving fast learning aided by the potentially suboptimal baseline policy.

**f-CPO and f-PCPO.** f-CPO and f-PCPO fail to improve the reward and have more cost violations. Most likely this is due to persistent supervision from the baseline policy which need not satisfy the cost constraints nor have high reward. For example, in the car-racing task we observe that the value

of the divergence cost decreases throughout the training. This implies that the learned policy overly evolves to the sub-optimal baseline policy and hence degrades the reward performance.

**d-CPO and d-PCPO.** d-CPO and d-PCPO improve the reward slowly and have more cost violations. They do not use projection to quickly learn from $\pi_B$. For example, in the car-racing task $J_D(\pi)$ in d-CPO and d-PCPO are high compared to SPACE throughout the training. This suggests that simply adding an IL objective in the reward objective is susceptible to a suboptimal $\pi_B$. Hence these approaches require a good selection of the initial value of $\lambda^k$ and $\beta$.

**Suboptimal $\pi_B^{\text{cost}}$ and $\pi_B^{\text{reward}}$.** The learning curves of $J_C(\pi)$ and the $J_R(\pi)$ over policy updates are shown for the point gather task in Fig. 4. We use two *suboptimal* baseline policies: $\pi_B^{\text{cost}}$ and $\pi_B^{\text{reward}}$, and $h_C$ is set to 0.5. They do not solve the task at hand.

We observe that SPACE robustly satisfies the cost constraints in all cases even when learning from $\pi_B^{\text{reward}}$. In addition, we observe that learning guided by $\pi_B^{\text{reward}}$ achieves faster reward learning efficiency at the *initial* iteration. This is because $J_R(\pi_B^{\text{reward}}) > J_R(\pi_B^{\text{cost}})$ as seen in the reward plot. Furthermore, we observe that learning guided by $\pi_B^{\text{cost}}$ achieves faster reward learning efficiency at the *later* iteration. This is because that by starting in the interior of the cost constraint set (*i.e.,* $J_C(\pi_B^{\text{cost}}) \approx 0 \leq h_C$), the agent can safely exploit the baseline policy. The results show that SPACE enables fast convergence to a constraint-satisfying policy, even if $\pi_B$ does not meet the constraints itself or does not optimize the reward.

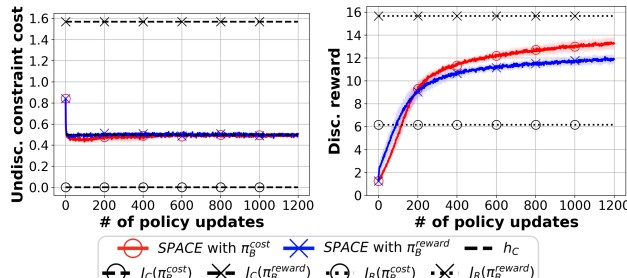

Figure 4: Learning from *suboptimal* $\pi_B$. The undiscounted constraint cost and the discounted reward over policy updates for the point gather task. The solid line is the mean and the shaded area is the standard deviation over 5 runs. The black dashed line is the cost constraint threshold $h_C$. We observe that SPACE satisfies the cost constraints even when learning from the suboptimal $\pi_B$. (Best viewed in color.)

## 7 CONCLUSION

We address the problem of learning constraint-satisfying policies given a baseline policy from either a teacher agent or demonstration data. The proposed algorithm effectively learns from the potentially suboptimal baseline policy without violating the constraints. SPACE achieves superior reward and cost performance compared with state-of-the-art approaches (*i.e.,* PCPO). We further analyze the convergence of SPACE and provide an effective approach to controlling the distance between the learned and baseline policies. Future work will consider learning system dynamics to account for the uncertainty of the environment and hence enable safe learning in real applications.

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

# Supplementary Material for
# Accelerating Safe Reinforcement Learning
# with Constraint-mismatched Policies

**Outline.** Supplementary material is outlined as follows. Section A discusses the impact of the proposed algorithm. Section B details the proof of updating $h_D$ in Lemma 4.1. Section C describes the proof of analytical solution to SPACE in Eq. (10). Section D gives the proof of finite-time guarantee of SPACE in Theorem 5.1 and discuss the difference between the KL-divergence and 2-norm projections. Section E assembles the additional experiment results to provide a detailed examination of the proposed algorithm compared to the baselines. These include:

- evaluation of the discounted reward versus the cumulative undiscounted constraint cost to demonstrate that SPACE achieves better reward performance with fewer cost constraint violations,

- evaluation of performance of SPACE guided by baseline policies with different $J_C(\pi_B)$ to demonstrate that SPACE safely learns from the baseline policies which need not satisfy the cost constraint,

- ablation studies of using a fixed $h_D$ in SPACE to demonstrate the importance of using the dynamic $h_D$ to improve the reward and cost performance,

- comparison of SPACE and other annealing approaches to demonstrate that SPACE exploits the baseline policy effectively,

- comparison of SPACE under the KL-divergence and the 2-norm projections to demonstrate that they converge to different stationary points,

- evaluation of using different initial values of $h_D^0$ to demonstrate that the selection of the initial value does not affect the performance of SPACE drastically.

Section E also details the environment parameters, the architectures of policies, computational cost, infrastructure for computation and the instructions for executing the code. Section F provides a procedure for getting a baseline human policy. Finally, we fill the Machine Learning Reproducibility Checklist in Section G.

## A    IMPACT OF SPACE

Many autonomous systems such as self-driving cars and autonomous robots are complex. In order to deal with this complexity, researchers are increasingly using reinforcement learning in conjunction with imitation learning for designing control policies. The more we can learn from a previous policy (*e.g.,* human demonstration, previous applications), the fewer resources (*e.g.,* time, energy, engineering effort, cost) we need to learn a new policy. The proposed algorithm could be applied in many fields where learning a policy can take advantage of prior applications while providing assurances for the consideration of fairness, safety, or other costs. For example, in a dialogue system where an agent is intended to converse with a human, the agent should safely learn from human preferences while avoiding producing biased or offensive responses. In addition, in the self-driving car domain where an agent learns a driving policy, the agent should safely learn from human drivers while avoiding a crash. Moreover, in the personalized robotic assistant setting where an agent learns from human demonstration, the agent should carefully imitate humans without damaging itself or causing harm to nearby humans. These examples highlight the potential impact of the proposed algorithm for accelerating safe reinforcement learning by adapting prior knowledge. This can open the door to advances in lifelong learning and adaptation of agents to different contexts.

One deficiency of the proposed algorithm is that the agent still experiments with cost constraint violation when learning control policies. This is because that any learning-based system needs to experiment with various actions to find a constraint-satisfying policy. Even though the agent does not violate the safety constraints during the learning phase, any change or perturbation of the environment that was not envisioned at the time of programming or training may lead to a catastrophic failure during run-time. These systems cannot guarantee that sensor inputs will not induce undesirable

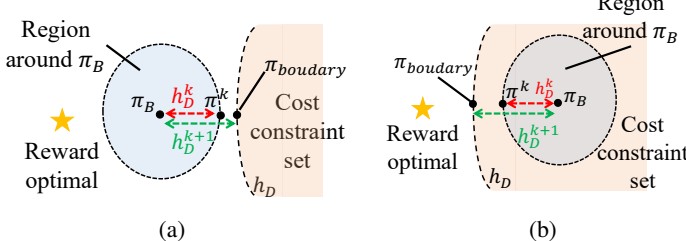

Figure 5: **(a)** Illustrating when $\pi_B$ is *outside* the cost constraint set. **(b)** Illustrating when $\pi_B$ is *inside* the cost constraint set. $\pi_{boundary}$ is the policy with $J_C(\pi_{boundary}) = h_C$. We aim to bound $h_D^{k+1}$ (*i.e.*, the KL-divergence between $\pi_{boundary}$ and $\pi_B$) by using $h_D^k$.

consequences, nor can the systems adapt and support safety in situations in which new objectives are created. This creates huge concerns in safety-critical applications such as self-driving vehicles and personalized chatbot system.

This raises several questions: What human-agent communication is needed to bring humans in the loop to increase safety guarantees for the autonomous system? How can trust and safety constraints be incorporated into the planning and control processes? How can one effectively identify unsafe plans of the baseline policy? We believe this paper will encourage future work to develop rigorous design and analysis tools for continual safety assurance in conjunction with using baseline policies from previous applications.

## B    PROOF OF UPDATING $h_D$ IN LEMMA 4.1

*Proof.* Based on Theorem 1 in Achiam et al. (2017), for any two policies $\pi$ and $\pi'$ we have

$$J_C(\pi') - J_C(\pi) \geq \frac{1}{1-\gamma} \mathbb{E}_{\substack{s \sim d^\pi \\ a \sim \pi'}} \left[ A_C^\pi(s,a) - \frac{2\gamma\epsilon_C^{\pi'}}{1-\gamma} \sqrt{\frac{1}{2} D_{\mathrm{KL}}(\pi'(s)||\pi(s))} \right]$$

$$\Rightarrow \quad \frac{2\gamma\epsilon_C^{\pi'}}{(1-\gamma)^2} \mathbb{E}_{s \sim d^\pi} \left[ \sqrt{\frac{1}{2} D_{\mathrm{KL}}(\pi'(s)||\pi(s))} \right] \geq -J_C(\pi') + J_C(\pi) + \frac{1}{1-\gamma} \mathbb{E}_{\substack{s \sim d^\pi \\ a \sim \pi'}} \left[ A_C^\pi(s,a) \right]$$

$$\Rightarrow \quad \frac{2\gamma\epsilon_C^{\pi'}}{(1-\gamma)^2} \mathbb{E}_{s \sim d^\pi} \left[ \sqrt{\frac{1}{2} D_{\mathrm{KL}}(\pi'(s)||\pi(s))} \right] \geq -J_C(\pi') + J_C(\pi)$$

$$\Rightarrow \quad \frac{\sqrt{2}\gamma\epsilon_C^{\pi'}}{(1-\gamma)^2} \sqrt{\mathbb{E}_{s \sim d^\pi} \left[ D_{\mathrm{KL}}(\pi'(s)||\pi(s)) \right]} \geq -J_C(\pi') + J_C(\pi)$$

$$\Rightarrow \quad \mathbb{E}_{s \sim d^\pi} \left[ D_{\mathrm{KL}}(\pi'(s)||\pi(s)) \right] \geq \frac{(1-\gamma)^4 (-J_C(\pi') + J_C(\pi))^2}{2\gamma^2 \epsilon_C^{\pi'^2}}. \tag{16}$$

The fourth inequality follows from Jensen's inequality. We then define $\varphi(\pi(s)) \doteq \sum_i \pi(a(i)|s) \log \pi(a(i)|s)$. By Three-point Lemma (Chen & Teboulle, 1993), for any three policies $\pi, \pi'$, and $\hat{\pi}$ we have

$$\mathbb{E}_{s \sim d^\pi} \left[ D_{\mathrm{KL}}(\pi'(s)||\hat{\pi}(s)) \right] = \mathbb{E}_{s \sim d^\pi} \left[ D_{\mathrm{KL}}(\pi'(s)||\pi(s)) \right] + \mathbb{E}_{s \sim d^\pi} \left[ D_{\mathrm{KL}}(\pi(s)||\hat{\pi}(s)) \right]$$
$$-\mathbb{E}_{s \sim d^\pi} \left[ (\nabla\varphi(\hat{\pi}(s)) - \nabla\varphi(\pi(s)))^T (\pi'(s) - \pi(s)) \right]. \tag{17}$$

Let $\pi_{boundary}$ denote a policy satisfying $J_C(\pi_{boundary}) = h_C$ (*i.e.*, $\pi_{boundary}$ is in the boundary of the set of the policies which satisfy the cost constraint $J_C(\pi) \leq h_C$). Let $\pi' = \pi_{boundary}, \hat{\pi} = \pi_B$

and $\pi = \pi^k$ in Eq. (16) and Eq. (17) (this is illustrated in Fig. 5). Then we have

$$
\begin{aligned}
&\mathbb{E}_{s\sim d^{\pi^k}}\left[D_{\mathrm{KL}}(\pi_{boundary}(s)||\pi_B(s))\right] - \mathbb{E}_{s\sim d^{\pi^k}}\left[D_{\mathrm{KL}}(\pi^k(s)||\pi_B(s))\right] \\
&= \mathbb{E}_{s\sim d^{\pi^k}}\left[D_{\mathrm{KL}}(\pi_{boundary}(s)||\pi^k(s))\right] \\
&\qquad - \mathbb{E}_{s\sim d^{\pi^k}}\left[(\nabla\varphi(\pi_B(s)) - \nabla\varphi(\pi^k(s)))^T(\pi_{boundary}(s) - \pi^k(s))\right] \\
&\geq \frac{(1-\gamma)^4(-J_C(\pi_{boundary}) + J_C(\pi^k))^2}{2\gamma^2\epsilon_C^{\pi'\,2}} \\
&\qquad - \mathbb{E}_{s\sim d^{\pi^k}}\left[(\nabla\varphi(\pi_B(s)) - \nabla\varphi(\pi^k(s)))^T(\pi_{boundary}(s) - \pi^k(s))\right] \\
&= \frac{(1-\gamma)^4(-h_C + J_C(\pi^k))^2}{2\gamma^2\epsilon_C^{\pi'\,2}} \\
&\qquad - \mathbb{E}_{s\sim d^{\pi^k}}\left[(\nabla\varphi(\pi_B(s)) - \nabla\varphi(\pi^k(s)))^T(\pi_{boundary}(s) - \pi^k(s))\right] \\
&= \mathcal{O}\Big(\big(-h_C + J_C(\pi^k)\big)^2\Big),
\end{aligned}
\tag{18}
$$

where $J_C(\pi_{boundary}) = h_C$.

For the first case in Fig. 5(a), we would like to have $\mathcal{U}_1 \cap \mathcal{U}_2^{k+1} \neq \emptyset$ (feasibility). For the second case in Fig. 5(b), we would like to have $\mathcal{U}_2^{k+1} \cap \partial\mathcal{U}_1 \neq \emptyset$ (exploration). These implies that the policy in step $k + 1$ is $\pi_{boundary}$ which satisfies $\mathcal{U}_1 \cap \mathcal{U}_2^{k+1} \neq \emptyset$ and $\mathcal{U}_2^{k+1} \cap \partial\mathcal{U}_1 \neq \emptyset$.

Now let $h_D^{k+1} \doteq \mathbb{E}_{s\sim d^{\pi^k}}\left[D_{\mathrm{KL}}(\pi_{boundary}(s)||\pi_B(s))\right]$ and $h_D^k \doteq \mathbb{E}_{s\sim d^{\pi^k}}[D_{\mathrm{KL}}(\pi^k(s)||\pi_B(s))]$. Then Eq. 18 implies

$$
h_D^{k+1} \geq \mathcal{O}\Big(\big(-h_C + J_C(\pi^k)\big)^2\Big) + h_D^k.
$$

$\square$

Lemma 4.1 theoretically ensures $h_D$ is large enough to guarantee feasibility and exploration of the agent. Note that we do not provide guarantees for finding an optimal policy. This requires additional assumptions on the objective function (*e.g.,* convexity).

In addition, the goal of this paper is to understand and analyze how to effectively exploit a baseline policy in constrained RL. Without such an analysis, we are not confident in deploying SPACE in real applications. Furthermore, the question of safely using baseline policies has a practical potential. It is less studied by prior work (Achiam et al., 2017; Chow et al., 2019; Tessler et al., 2018; Yang et al., 2020).

## C  Proof of Analytical Solution to SPACE in Eq. (10)

We first approximate the three stages in SPACE using the following approximations.

**Step 1.** Approximating Eq. (4) yields

$$
\boldsymbol{\theta}^{k+\frac{1}{3}} = \arg\max_{\boldsymbol{\theta}} \boldsymbol{g}^{k\,T}(\boldsymbol{\theta} - \boldsymbol{\theta}^k) \quad \text{s.t.} \ \frac{1}{2}(\boldsymbol{\theta} - \boldsymbol{\theta}^k)^T\boldsymbol{F}^k(\boldsymbol{\theta} - \boldsymbol{\theta}^k) \leq \delta.
\tag{19}
$$

**Step 2 and Step 3.** Approximating Eq. (5) and (6), similarly yields

$$
\boldsymbol{\theta}^{k+\frac{2}{3}} = \arg\min_{\boldsymbol{\theta}} \frac{1}{2}(\boldsymbol{\theta} - \boldsymbol{\theta}^{k+\frac{1}{3}})^T\boldsymbol{L}(\boldsymbol{\theta} - \boldsymbol{\theta}^{k+\frac{1}{3}}) \quad \text{s.t.} \ \boldsymbol{a}^{k\,T}(\boldsymbol{\theta} - \boldsymbol{\theta}^k) + b^k \leq 0,
\tag{20}
$$

$$
\boldsymbol{\theta}^{k+1} = \arg\min_{\boldsymbol{\theta}} \frac{1}{2}(\boldsymbol{\theta} - \boldsymbol{\theta}^{k+\frac{2}{3}})^T\boldsymbol{L}(\boldsymbol{\theta} - \boldsymbol{\theta}^{k+\frac{2}{3}}) \quad \text{s.t.} \ \boldsymbol{c}^{k\,T}(\boldsymbol{\theta} - \boldsymbol{\theta}^k) + d^k \leq 0,
\tag{21}
$$

where $\boldsymbol{L} = \boldsymbol{I}$ for the 2-norm projection and $\boldsymbol{L} = \boldsymbol{F}^k$ for the KL-divergence projection.

*Proof.* For the first problem in Eq. (19), since $\boldsymbol{F}^k$ is the Fisher Information matrix, it is positive semi-definite. Hence it is a convex program with quadratic inequality constraints. If the primal problem has a feasible point, then Slater's condition is satisfied and strong duality holds. Let $\boldsymbol{\theta}^*$ and $\lambda^*$ denote the solutions to the primal and dual problems, respectively. In addition, the primal objective function is continuously differentiable. Hence the Karush-Kuhn-Tucker (KKT) conditions are necessary and sufficient for the optimality of $\boldsymbol{\theta}^*$ and $\lambda^*$. We now form the Lagrangian:

$$\mathcal{L}(\boldsymbol{\theta}, \lambda) = -\boldsymbol{g}^{k^T}(\boldsymbol{\theta} - \boldsymbol{\theta}^k) + \lambda\Big(\frac{1}{2}(\boldsymbol{\theta} - \boldsymbol{\theta}^k)^T \boldsymbol{F}^k(\boldsymbol{\theta} - \boldsymbol{\theta}^k) - \delta\Big).$$

And we have the following KKT conditions:

$$-\boldsymbol{g}^k + \lambda^* \boldsymbol{F}^k \boldsymbol{\theta}^* - \lambda^* \boldsymbol{F}^k \boldsymbol{\theta}^k = 0 \qquad \nabla_{\boldsymbol{\theta}} \mathcal{L}(\boldsymbol{\theta}^*, \lambda^*) = 0 \tag{22}$$

$$\frac{1}{2}(\boldsymbol{\theta}^* - \boldsymbol{\theta}^k)^T \boldsymbol{F}^k(\boldsymbol{\theta}^* - \boldsymbol{\theta}^k) - \delta = 0 \qquad \nabla_{\lambda} \mathcal{L}(\boldsymbol{\theta}^*, \lambda^*) = 0 \tag{23}$$

$$\frac{1}{2}(\boldsymbol{\theta}^* - \boldsymbol{\theta}^k)^T \boldsymbol{F}^k(\boldsymbol{\theta}^* - \boldsymbol{\theta}^k) - \delta \leq 0 \qquad \text{primal constraints} \tag{24}$$

$$\lambda^* \geq 0 \qquad \text{dual constraints} \tag{25}$$

$$\lambda^*\Big(\frac{1}{2}(\boldsymbol{\theta}^* - \boldsymbol{\theta}^k)^T \boldsymbol{F}^k(\boldsymbol{\theta}^* - \boldsymbol{\theta}^k) - \delta\Big) = 0 \qquad \text{complementary slackness} \tag{26}$$

By Eq. (22), we have $\boldsymbol{\theta}^* = \boldsymbol{\theta}^k + \frac{1}{\lambda^*}\boldsymbol{F}^{k^{-1}}\boldsymbol{g}^k$. And by plugging Eq. (22) into Eq. (23), we have $\lambda^* = \sqrt{\frac{\boldsymbol{g}^{k^T}\boldsymbol{F}^{k^{-1}}\boldsymbol{g}^k}{2\delta}}$. Hence we have a solution

$$\boldsymbol{\theta}^{k+\frac{1}{3}} = \boldsymbol{\theta}^* = \boldsymbol{\theta}^k + \sqrt{\frac{2\delta}{\boldsymbol{g}^{k^T}\boldsymbol{F}^{k^{-1}}\boldsymbol{g}^k}}\boldsymbol{F}^{k^{-1}}\boldsymbol{g}^k, \tag{27}$$

which also satisfies Eq. (24), Eq. (25), and Eq. (26).

For the second problem in Eq. (20), we follow the same procedure for the first problem to form the Lagrangian:

$$\mathcal{L}(\boldsymbol{\theta}, \lambda) = \frac{1}{2}(\boldsymbol{\theta} - \boldsymbol{\theta}^{k+\frac{1}{3}})^T \boldsymbol{L}(\boldsymbol{\theta} - \boldsymbol{\theta}^{k+\frac{1}{3}}) + \lambda(\boldsymbol{a}^{k^T}(\boldsymbol{\theta} - \boldsymbol{\theta}^k) + b^k).$$

And we have the following KKT conditions:

$$\boldsymbol{L}\boldsymbol{\theta}^* - \boldsymbol{L}\boldsymbol{\theta}^{k+\frac{1}{3}} + \lambda^* \boldsymbol{a}^k = 0 \qquad \nabla_{\boldsymbol{\theta}} \mathcal{L}(\boldsymbol{\theta}^*, \lambda^*) = 0 \tag{28}$$

$$\boldsymbol{a}^{k^T}(\boldsymbol{\theta}^* - \boldsymbol{\theta}^k) + b^k = 0 \qquad \nabla_{\lambda} \mathcal{L}(\boldsymbol{\theta}^*, \lambda^*) = 0 \tag{29}$$

$$\boldsymbol{a}^{k^T}(\boldsymbol{\theta}^* - \boldsymbol{\theta}^k) + b^k \leq 0 \qquad \text{primal constraints} \tag{30}$$

$$\lambda^* \geq 0 \qquad \text{dual constraints} \tag{31}$$

$$\lambda^*(\boldsymbol{a}^{k^T}(\boldsymbol{\theta}^* - \boldsymbol{\theta}^k) + b^k) = 0 \qquad \text{complementary slackness} \tag{32}$$

By Eq. (28), we have $\boldsymbol{\theta}^* = \boldsymbol{\theta}^k + \lambda^* \boldsymbol{L}^{-1}\boldsymbol{a}^k$. And by plugging Eq. (28) into Eq. (29) and Eq. (31), we have $\lambda^* = \max(0, \frac{\boldsymbol{a}^{k^T}(\boldsymbol{\theta}^{k+\frac{1}{3}} - \boldsymbol{\theta}^k) + b^k}{\boldsymbol{a}^k \boldsymbol{L}^{-1}\boldsymbol{a}^k})$. Hence we have a solution

$$\boldsymbol{\theta}^{k+\frac{2}{3}} = \boldsymbol{\theta}^* = \boldsymbol{\theta}^{k+\frac{1}{3}} - \max(0, \frac{\boldsymbol{a}^{k^T}(\boldsymbol{\theta}^{k+\frac{1}{3}} - \boldsymbol{\theta}^k) + b^k}{\boldsymbol{a}^{k^T}\boldsymbol{L}^{-1}\boldsymbol{a}^{k^T}})\boldsymbol{L}^{-1}\boldsymbol{a}^k, \tag{33}$$

which also satisfies Eq. (30) and Eq. (32).

For the third problem in Eq. (21), instead of doing the projection on $\pi^{k+\frac{2}{3}}$ which is the intermediate policy obtained in the second step, we project the policy $\pi^{k+\frac{1}{3}}$ onto the cost constraint. This allows us to compute the projection without too much computational cost. We follow the same procedure for the first and second problems to form the Lagrangian:

$$\mathcal{L}(\boldsymbol{\theta}, \lambda) = \frac{1}{2}(\boldsymbol{\theta} - \boldsymbol{\theta}^{k+\frac{1}{3}})^T \boldsymbol{L}(\boldsymbol{\theta} - \boldsymbol{\theta}^{k+\frac{1}{3}}) + \lambda(\boldsymbol{c}^{k^T}(\boldsymbol{\theta} - \boldsymbol{\theta}^k) + d^k).$$

And we have the following KKT conditions:

$$\boldsymbol{L\theta}^* - \boldsymbol{L\theta}^{k+\frac{1}{3}} + \lambda^* \boldsymbol{c}^k = 0 \qquad \nabla_{\boldsymbol{\theta}}\mathcal{L}(\boldsymbol{\theta}^*, \lambda^*) = 0 \tag{34}$$

$$\boldsymbol{c}^{k^T}(\boldsymbol{\theta}^* - \boldsymbol{\theta}^k) + d^k = 0 \qquad \nabla_{\lambda}\mathcal{L}(\boldsymbol{\theta}^*, \lambda^*) = 0 \tag{35}$$

$$\boldsymbol{c}^{k^T}(\boldsymbol{\theta}^* - \boldsymbol{\theta}^k) + d^k \leq 0 \qquad \text{primal constraints} \tag{36}$$

$$\lambda^* \geq 0 \qquad \text{dual constraints} \tag{37}$$

$$\lambda^*(\boldsymbol{c}^{k^T}(\boldsymbol{\theta}^* - \boldsymbol{\theta}^k) + d^k) = 0 \qquad \text{complementary slackness} \tag{38}$$

By Eq. (34), we have $\boldsymbol{\theta}^* = \boldsymbol{\theta}^k + \lambda^* \boldsymbol{L}^{-1}\boldsymbol{c}^k$. And by plugging Eq. (34) into Eq. (35) and Eq. (37), we have $\lambda^* = \max(0, \frac{\boldsymbol{c}^{k^T}(\boldsymbol{\theta}^{k+\frac{1}{3}} - \boldsymbol{\theta}^k) + d^k}{\boldsymbol{c}^k \boldsymbol{L}^{-1}\boldsymbol{c}^k})$. Hence we have a solution

$$\boldsymbol{\theta}^{k+1} = \boldsymbol{\theta}^* = \boldsymbol{\theta}^{k+\frac{1}{3}} - \max(0, \frac{\boldsymbol{c}^{k^T}(\boldsymbol{\theta}^{k+\frac{1}{3}} - \boldsymbol{\theta}^k) + d^k}{\boldsymbol{c}^{k^T}\boldsymbol{L}^{-1}\boldsymbol{c}^{k^T}})\boldsymbol{L}^{-1}\boldsymbol{c}^k. \tag{39}$$

Hence by combining Eq. (27), Eq. (33) and Eq. (39), we have

$$\boldsymbol{\theta}^{k+1} = \boldsymbol{\theta}^k + \sqrt{\frac{2\delta}{\boldsymbol{g}^{k^T}\boldsymbol{F}^{k^{-1}}\boldsymbol{g}^k}}\boldsymbol{F}^{k^{-1}}\boldsymbol{g}^k - \max(0, \frac{\sqrt{\frac{2\delta}{\boldsymbol{g}^{k^T}\boldsymbol{F}^{k^{-1}}\boldsymbol{g}^k}}\boldsymbol{a}^{k^T}\boldsymbol{F}^{k^{-1}}\boldsymbol{g}^k + b^k}{\boldsymbol{a}^{k^T}\boldsymbol{L}^{-1}\boldsymbol{a}^k})\boldsymbol{L}^{-1}\boldsymbol{a}^k$$

$$- \max(0, \frac{\sqrt{\frac{2\delta}{\boldsymbol{g}^{k^T}\boldsymbol{F}^{k^{-1}}\boldsymbol{g}^k}}\boldsymbol{c}^{k^T}\boldsymbol{F}^{k^{-1}}\boldsymbol{g}^k + d^k}{\boldsymbol{c}^{k^T}\boldsymbol{L}^{-1}\boldsymbol{c}^k})\boldsymbol{L}^{-1}\boldsymbol{c}^k.$$

$\square$

# D   PROOF OF FINITE-TIME GUARANTEE OF SPACE IN THEOREM 5.1

We now describe the reason for choosing two variants of $\epsilon$-FOSP under two possible projections. Let $\eta_R^k$ denote the step size for the reward, $\eta_D^k$ denote the step size for the divergence cost, and $\eta_C^k$ denote the step size for the constraint cost. Without loss of generality, under the KL-divergence projection, at step $k + 1$ SPACE does

$$\boldsymbol{\theta}^{k+1} = \boldsymbol{\theta}^k + \eta_R^k \boldsymbol{F}^{k^{-1}}\boldsymbol{g}^k - \eta_D^k \boldsymbol{F}^{k^{-1}}\boldsymbol{a}^k - \eta_C^k \boldsymbol{F}^{k^{-1}}\boldsymbol{c}^k.$$

Similarly, under the 2-norm projection, at step $k + 1$ SPACE does

$$\boldsymbol{\theta}^{k+1} = \boldsymbol{\theta}^k + \eta_R^k \boldsymbol{F}^k \boldsymbol{g}^k - \eta_D^k \boldsymbol{a}^k - \eta_C^k \boldsymbol{c}^k.$$

With this definition, we have the following Lemma.

**Lemma D.1** (**Stationary Points for SPACE**). *Under the KL-divergence projection, SPACE converges to a stationary point $\boldsymbol{\theta}^*$ satisfying*

$$\eta_R^* \boldsymbol{g}^* = \eta_D^* \boldsymbol{a}^* + \eta_C^* \boldsymbol{c}^*.$$

*Under the 2-norm projection, SPACE converges to a stationary point $\boldsymbol{\theta}^*$ satisfying*

$$\eta_R^* \boldsymbol{g}^* = \boldsymbol{F}^*(\eta_D^* \boldsymbol{a}^* + \eta_C^* \boldsymbol{c}^*).$$

*Proof.* Under the KL-divergence projection, by using the definition of a stationary point we have

$$\boldsymbol{\theta}^* = \boldsymbol{\theta}^* + \eta_R^* \boldsymbol{F}^{*^{-1}}\boldsymbol{g}^* - \eta_D^* \boldsymbol{F}^{*^{-1}}\boldsymbol{a}^* - \eta_C^* \boldsymbol{F}^{*^{-1}}\boldsymbol{c}^*$$

$$\Rightarrow \quad \eta_R^* \boldsymbol{F}^{*^{-1}}\boldsymbol{g}^* = \eta_D^* \boldsymbol{F}^{*^{-1}}\boldsymbol{a}^* + \eta_C^* \boldsymbol{F}^{*^{-1}}\boldsymbol{c}^*$$

$$\Rightarrow \quad \eta_R^* \boldsymbol{g}^* = \eta_D^* \boldsymbol{a}^* + \eta_C^* \boldsymbol{c}^*.$$

Under the 2-norm projection, by using the definition of a stationary point we have

$$\boldsymbol{\theta}^* = \boldsymbol{\theta}^* + \eta_R^* \boldsymbol{F}^{*^{-1}}\boldsymbol{g}^* - \eta_D^* \boldsymbol{a}^* - \eta_C^* \boldsymbol{c}^*$$

$$\Rightarrow \quad \eta_R^* \boldsymbol{F}^{*^{-1}}\boldsymbol{g}^* = \eta_D^* \boldsymbol{a}^* + \eta_C^* \boldsymbol{c}^*$$

$$\Rightarrow \quad \eta_R^* \boldsymbol{g}^* = \boldsymbol{F}^*(\eta_D^* \boldsymbol{a}^* + \eta_C^* \boldsymbol{c}^*).$$

$\square$

Hence Lemma D.1 motivates the need for defining two variants of FOSP.

Before proving Theorem 5.1, we need the following Lemma. Define $\mathcal{P}_{\mathcal{C}}^{\boldsymbol{L}}(\boldsymbol{\theta}) \doteq \underset{\boldsymbol{\theta}' \in \mathcal{C}}{\arg\min} \|\boldsymbol{\theta} - \boldsymbol{\theta}'\|_{\boldsymbol{L}}^2 = \underset{\boldsymbol{\theta}' \in \mathcal{C}}{\arg\min} (\boldsymbol{\theta} - \boldsymbol{\theta}')^T \boldsymbol{L} (\boldsymbol{\theta} - \boldsymbol{\theta}')$, and $\boldsymbol{L} = \boldsymbol{F}^k$ under the KL-divergence projection, and $\boldsymbol{L} = \boldsymbol{I}$ under the 2-norm projection.

**Lemma D.2 (Contraction of Projection** (Yang et al., 2020)**).** *For any $\boldsymbol{\theta}$, $\boldsymbol{\theta}^* = \mathcal{P}_{\mathcal{C}}^{\boldsymbol{L}}(\boldsymbol{\theta})$ if and only if $(\boldsymbol{\theta} - \boldsymbol{\theta}^*)^T \boldsymbol{L} (\boldsymbol{\theta}' - \boldsymbol{\theta}^*) \leq 0, \forall \boldsymbol{\theta}' \in \mathcal{C}$.*

*Proof.* ($\Rightarrow$) Let $\boldsymbol{\theta}^* = \mathcal{P}_{\mathcal{C}}^{\boldsymbol{L}}(\boldsymbol{\theta})$ for a given $\boldsymbol{\theta} \notin \mathcal{C}$, $\boldsymbol{\theta}' \in \mathcal{C}$ be such that $\boldsymbol{\theta}' \neq \boldsymbol{\theta}^*$, and $\alpha \in (0, 1)$. Then we have

$$
\begin{aligned}
\|\boldsymbol{\theta} - \boldsymbol{\theta}^*\|_{\boldsymbol{L}}^2 &\leq \|\boldsymbol{\theta} - (\boldsymbol{\theta}^* + \alpha(\boldsymbol{\theta}' - \boldsymbol{\theta}^*))\|_{\boldsymbol{L}}^2 \\
&= \|\boldsymbol{\theta} - \boldsymbol{\theta}^*\|_{\boldsymbol{L}}^2 + \alpha^2 \|\boldsymbol{\theta}' - \boldsymbol{\theta}^*\|_{\boldsymbol{L}}^2 - 2\alpha(\boldsymbol{\theta} - \boldsymbol{\theta}^*)^T \boldsymbol{L}(\boldsymbol{\theta}' - \boldsymbol{\theta}^*) \\
\Rightarrow (\boldsymbol{\theta} - \boldsymbol{\theta}^*)^T \boldsymbol{L}(\boldsymbol{\theta}' - \boldsymbol{\theta}^*) &\leq \frac{\alpha}{2} \|\boldsymbol{\theta}' - \boldsymbol{\theta}^*\|_{\boldsymbol{L}}^2.
\end{aligned}
\tag{40}
$$

Since the right hand side of Eq. (40) can be made arbitrarily small for a given $\alpha$, we have

$$
(\boldsymbol{\theta} - \boldsymbol{\theta}^*)^T \boldsymbol{L}(\boldsymbol{\theta}' - \boldsymbol{\theta}^*) \leq 0, \forall \boldsymbol{\theta}' \in \mathcal{C}.
$$

($\Leftarrow$) Let $\boldsymbol{\theta}^* \in \mathcal{C}$ be such that $(\boldsymbol{\theta} - \boldsymbol{\theta}^*)^T \boldsymbol{L}(\boldsymbol{\theta}' - \boldsymbol{\theta}^*) \leq 0, \forall \boldsymbol{\theta}' \in \mathcal{C}$. We show that $\boldsymbol{\theta}^*$ must be the optimal solution. Let $\boldsymbol{\theta}' \in \mathcal{C}$ and $\boldsymbol{\theta}' \neq \boldsymbol{\theta}^*$. Then we have

$$
\begin{aligned}
\|\boldsymbol{\theta} - \boldsymbol{\theta}'\|_{\boldsymbol{L}}^2 - \|\boldsymbol{\theta} - \boldsymbol{\theta}^*\|_{\boldsymbol{L}}^2 &= \|\boldsymbol{\theta} - \boldsymbol{\theta}^* + \boldsymbol{\theta}^* - \boldsymbol{\theta}'\|_{\boldsymbol{L}}^2 - \|\boldsymbol{\theta} - \boldsymbol{\theta}^*\|_{\boldsymbol{L}}^2 \\
&= \|\boldsymbol{\theta} - \boldsymbol{\theta}^*\|_{\boldsymbol{L}}^2 + \|\boldsymbol{\theta}' - \boldsymbol{\theta}^*\|_{\boldsymbol{L}}^2 - 2(\boldsymbol{\theta} - \boldsymbol{\theta}^*)^T \boldsymbol{L}(\boldsymbol{\theta}' - \boldsymbol{\theta}^*) - \|\boldsymbol{\theta} - \boldsymbol{\theta}^*\|_{\boldsymbol{L}}^2 \\
&> 0 \\
\Rightarrow \|\boldsymbol{\theta} - \boldsymbol{\theta}'\|_{\boldsymbol{L}}^2 &> \|\boldsymbol{\theta} - \boldsymbol{\theta}^*\|_{\boldsymbol{L}}^2.
\end{aligned}
$$

Hence, $\boldsymbol{\theta}^*$ is the optimal solution to the optimization problem, and $\boldsymbol{\theta}^* = \mathcal{P}_{\mathcal{C}}^{\boldsymbol{L}}(\boldsymbol{\theta})$. $\qquad\square$

We now prove Theorem 5.1. Without loss of generality, on each learning episode SPACE updates the reward followed by the alternation of two projections onto the constraint sets (region around $\pi_B$ and the cost constraint set):

$$
\begin{aligned}
\boldsymbol{\theta}^{k+\frac{1}{3}} &= \boldsymbol{\theta}^k - \eta^k \boldsymbol{F}^{-1} \nabla f(\boldsymbol{\theta}^k), \ \boldsymbol{\theta}^{k+\frac{2}{3}} = \mathcal{P}_{\mathcal{C}_2}(\boldsymbol{\theta}^{k+\frac{1}{3}}), \ \boldsymbol{\theta}^{k+1} = \mathcal{P}_{\mathcal{C}_1}(\boldsymbol{\theta}^{k+\frac{2}{3}}), \text{if } \boldsymbol{\theta}^k \in \mathcal{C}_2, \\
\boldsymbol{\theta}^{k+\frac{1}{3}} &= \boldsymbol{\theta}^k - \eta^k \boldsymbol{F}^{-1} \nabla f(\boldsymbol{\theta}^k), \ \boldsymbol{\theta}^{k+\frac{2}{3}} = \mathcal{P}_{\mathcal{C}_1}(\boldsymbol{\theta}^{k+\frac{1}{3}}), \ \boldsymbol{\theta}^{k+1} = \mathcal{P}_{\mathcal{C}_2}(\boldsymbol{\theta}^{k+\frac{2}{3}}), \text{if } \boldsymbol{\theta}^k \in \mathcal{C}_1,
\end{aligned}
$$

where $\eta^k$ is the step size at step $k$.

*Proof.* **SPACE under the KL-divergence projection converges to an $\epsilon$-FOSP.** Based on Lemma D.2 under the KL-divergence projection, and setting $\boldsymbol{\theta} = \boldsymbol{\theta}^k - \eta^k \boldsymbol{F}^{k-1} \nabla f(\boldsymbol{\theta}^k)$, $\boldsymbol{\theta}^* = \boldsymbol{\theta}^{k+\frac{2}{3}}$ and $\boldsymbol{\theta}' = \boldsymbol{\theta}^k$, we have

$$
\begin{aligned}
(\boldsymbol{\theta}^k - \boldsymbol{\theta}^{k+\frac{2}{3}})^T \boldsymbol{F}^k (\boldsymbol{\theta}^k - \eta^k \boldsymbol{F}^{k-1} \nabla f(\boldsymbol{\theta}^k) - \boldsymbol{\theta}^{k+\frac{2}{3}}) &\leq 0 \\
\Rightarrow \quad \nabla f(\boldsymbol{\theta}^k)^T (\boldsymbol{\theta}^{k+\frac{2}{3}} - \boldsymbol{\theta}^k) &\leq -\frac{1}{\eta^k} (\boldsymbol{\theta}^{k+\frac{2}{3}} - \boldsymbol{\theta}^k)^T \boldsymbol{F}^k (\boldsymbol{\theta}^{k+\frac{2}{3}} - \boldsymbol{\theta}^k).
\end{aligned}
\tag{41}
$$

Based on the $L$-Lipschitz continuity of gradients and Eq. (41), we have

$$
\begin{aligned}
f(\boldsymbol{\theta}^{k+\frac{2}{3}}) &\leq f(\boldsymbol{\theta}^k) + \nabla f(\boldsymbol{\theta}^k)^T (\boldsymbol{\theta}^{k+\frac{2}{3}} - \boldsymbol{\theta}^k) + \frac{L}{2} \|\boldsymbol{\theta}^{k+\frac{2}{3}} - \boldsymbol{\theta}^k\|^2 \\
&\leq f(\boldsymbol{\theta}^k) - \frac{1}{\eta^k} (\boldsymbol{\theta}^{k+\frac{2}{3}} - \boldsymbol{\theta}^k)^T \boldsymbol{F}^k (\boldsymbol{\theta}^{k+\frac{2}{3}} - \boldsymbol{\theta}^k) + \frac{L}{2} \|\boldsymbol{\theta}^{k+\frac{2}{3}} - \boldsymbol{\theta}^k\|^2 \\
&= f(\boldsymbol{\theta}^k) - \frac{L}{2} \|\boldsymbol{\theta}^{k+\frac{2}{3}} - \boldsymbol{\theta}^k\|^2 - \nabla f(\boldsymbol{\theta}^{k+\frac{2}{3}})^T (\boldsymbol{\theta}^{k+1} - \boldsymbol{\theta}^{k+\frac{2}{3}}) - \frac{L}{2} \|\boldsymbol{\theta}^{k+1} - \boldsymbol{\theta}^{k+\frac{2}{3}}\|^2,
\end{aligned}
\tag{42}
$$

where the equality follows by setting $\delta$ (*i.e.,* the size of the trust region) such that

$$\eta^k = \frac{(\boldsymbol{\theta}^{k+\frac{2}{3}} - \boldsymbol{\theta}^k)^T \boldsymbol{F}^k (\boldsymbol{\theta}^{k+\frac{2}{3}} - \boldsymbol{\theta}^k)}{L\|\boldsymbol{\theta}^{k+\frac{2}{3}} - \boldsymbol{\theta}^k\|^2 + \nabla f(\boldsymbol{\theta}^{k+\frac{2}{3}})^T(\boldsymbol{\theta}^{k+1} - \boldsymbol{\theta}^{k+\frac{2}{3}}) + \frac{L}{2}\|\boldsymbol{\theta}^{k+1} - \boldsymbol{\theta}^{k+\frac{2}{3}}\|^2}.$$

Again, based on Lemma D.2, for $\boldsymbol{\theta} \in \mathcal{C}_2$ we have

$$
\begin{aligned}
& (\boldsymbol{\theta}^k - \eta^k \boldsymbol{F}^{k-1}\nabla f(\boldsymbol{\theta}^k) - \boldsymbol{\theta}^{k+\frac{2}{3}})\boldsymbol{F}^k(\boldsymbol{\theta} - \boldsymbol{\theta}^{k+\frac{2}{3}}) \leq 0 \\
\Rightarrow\quad & (-\eta^k \boldsymbol{F}^{k-1}\nabla f(\boldsymbol{\theta}^k))^T \boldsymbol{F}^k(\boldsymbol{\theta} - \boldsymbol{\theta}^{k+\frac{2}{3}}) \leq -(\boldsymbol{\theta}^k - \boldsymbol{\theta}^{k+\frac{2}{3}})^T \boldsymbol{F}^k(\boldsymbol{\theta} - \boldsymbol{\theta}^{k+\frac{2}{3}}) \\
\Rightarrow\quad & \nabla f(\boldsymbol{\theta}^k)^T(\boldsymbol{\theta} - \boldsymbol{\theta}^{k+\frac{2}{3}}) \geq \frac{1}{\eta^k}(\boldsymbol{\theta}^k - \boldsymbol{\theta}^{k+\frac{2}{3}})^T \boldsymbol{F}^k(\boldsymbol{\theta} - \boldsymbol{\theta}^{k+\frac{2}{3}}) \\
\Rightarrow\quad & \nabla f(\boldsymbol{\theta}^k)^T \boldsymbol{\theta} \geq \nabla f(\boldsymbol{\theta}^k)^T \boldsymbol{\theta}^{k+\frac{2}{3}} + \frac{1}{\eta^k}(\boldsymbol{\theta}^k - \boldsymbol{\theta}^{k+\frac{2}{3}})^T \boldsymbol{F}^k(\boldsymbol{\theta} - \boldsymbol{\theta}^{k+\frac{2}{3}}) \\
\Rightarrow\quad & f(\boldsymbol{\theta}^k)^T(\boldsymbol{\theta} - \boldsymbol{\theta}^k) \geq \nabla f(\boldsymbol{\theta}^k)^T(\boldsymbol{\theta}^{k+\frac{2}{3}} - \boldsymbol{\theta}^k) + \frac{1}{\eta^k}(\boldsymbol{\theta}^k - \boldsymbol{\theta}^{k+\frac{2}{3}})^T \boldsymbol{F}^k(\boldsymbol{\theta} - \boldsymbol{\theta}^{k+\frac{2}{3}}) \\
& \geq -\|\nabla f(\boldsymbol{\theta}^k)\|\|\boldsymbol{\theta}^{k+\frac{2}{3}} - \boldsymbol{\theta}^k\| - \frac{1}{\eta^k}\|\boldsymbol{\theta}^{k+\frac{2}{3}} - \boldsymbol{\theta}^k\|\|\boldsymbol{F}^k\|\|\boldsymbol{\theta} - \boldsymbol{\theta}^{k+\frac{2}{3}}\| \\
& \geq -\Big(G + \frac{D\sigma_1(\boldsymbol{F}^k)}{\eta^k}\Big)\|\boldsymbol{\theta}^{k+\frac{2}{3}} - \boldsymbol{\theta}^k\|,
\end{aligned}
\tag{43}
$$

where in the last two inequalities we use the property of the norm. Before reaching an $\epsilon$-FOSP, Eq. (43) implies that

$$
\begin{aligned}
& -\epsilon \geq \min_{\boldsymbol{\theta}\in\mathcal{C}_2} \nabla f(\boldsymbol{\theta}^k)^T(\boldsymbol{\theta} - \boldsymbol{\theta}^k) \geq -\Big(G + \frac{D\sigma_1(\boldsymbol{F}^k)}{\eta^k}\Big)\|\boldsymbol{\theta}^{k+\frac{2}{3}} - \boldsymbol{\theta}^k\| \\
\Rightarrow\quad & \|\boldsymbol{\theta}^{k+\frac{2}{3}} - \boldsymbol{\theta}^k\| \geq \frac{\epsilon}{G + \frac{D\sigma_1(\boldsymbol{F}^k)}{\eta^k}}.
\end{aligned}
\tag{44}
$$

Based on Eq. (42) and Eq. (44), we have

$$
\begin{aligned}
f(\boldsymbol{\theta}^{k+\frac{2}{3}}) &\leq f(\boldsymbol{\theta}^k) - \frac{L}{2}\|\boldsymbol{\theta}^{k+\frac{2}{3}} - \boldsymbol{\theta}^k\|^2 - \nabla f(\boldsymbol{\theta}^{k+\frac{2}{3}})^T(\boldsymbol{\theta}^{k+1} - \boldsymbol{\theta}^{k+\frac{2}{3}}) - \frac{L}{2}\|\boldsymbol{\theta}^{k+1} - \boldsymbol{\theta}^{k+\frac{2}{3}}\|^2 \\
&\leq f(\boldsymbol{\theta}^k) - \frac{L\epsilon^2}{2(G + \frac{D\sigma_1(\boldsymbol{F}^k)}{\eta^k})^2} - \nabla f(\boldsymbol{\theta}^{k+\frac{2}{3}})^T(\boldsymbol{\theta}^{k+1} - \boldsymbol{\theta}^{k+\frac{2}{3}}) - \frac{L}{2}\|\boldsymbol{\theta}^{k+1} - \boldsymbol{\theta}^{k+\frac{2}{3}}\|^2.
\end{aligned}
\tag{45}
$$

Based on the $L$-Lipschitz continuity of gradients, for the projection to the constraint set $\mathcal{C}_1$ we have

$$f(\boldsymbol{\theta}^{k+1}) \leq f(\boldsymbol{\theta}^{k+\frac{2}{3}}) + \nabla f(\boldsymbol{\theta}^{k+\frac{2}{3}})^T(\boldsymbol{\theta}^{k+1} - \boldsymbol{\theta}^{k+\frac{2}{3}}) + \frac{L}{2}\|\boldsymbol{\theta}^{k+1} - \boldsymbol{\theta}^{k+\frac{2}{3}}\|^2. \tag{46}$$

Combining Eq. (45) with Eq. (46), we have

$$f(\boldsymbol{\theta}^{k+1}) \leq f(\boldsymbol{\theta}^k) - \frac{L\epsilon^2}{2(G + \frac{D\sigma_1(\boldsymbol{F}^k)}{\eta^k})^2}. \tag{47}$$

Hence it takes $\mathcal{O}(\epsilon^{-2})$ iterations to reach an $\epsilon$-FOSP.

**SPACE under the 2-norm projection converges to an $\epsilon$-FOSP.** Based on Lemma D.2 under the 2-norm projection, and setting $\boldsymbol{\theta} = \boldsymbol{\theta}^k - \eta^k \boldsymbol{F}^{k-1}\nabla f(\boldsymbol{\theta}^k)$, $\boldsymbol{\theta}^* = \boldsymbol{\theta}^{k+\frac{2}{3}}$ and $\boldsymbol{\theta}' = \boldsymbol{\theta}^k$, we have

$$
\begin{aligned}
& (\boldsymbol{\theta}^k - \boldsymbol{\theta}^{k+\frac{2}{3}})^T(\boldsymbol{\theta}^k - \eta^k \boldsymbol{F}^{k-1}\nabla f(\boldsymbol{\theta}^k) - \boldsymbol{\theta}^{k+\frac{2}{3}}) \leq 0 \\
\Rightarrow & (\boldsymbol{F}^{k-1}\nabla f(\boldsymbol{\theta}^k))^T(\boldsymbol{\theta}^{k+\frac{2}{3}} - \boldsymbol{\theta}^k) \leq -\frac{1}{\eta^k}(\boldsymbol{\theta}^{k+\frac{2}{3}} - \boldsymbol{\theta}^k)^T(\boldsymbol{\theta}^{k+\frac{2}{3}} - \boldsymbol{\theta}^k).
\end{aligned}
\tag{48}
$$

Based on the $L$-Lipschitz continuity of gradients and Eq. (48), we have

$$
\begin{aligned}
f(\boldsymbol{\theta}^{k+\frac{2}{3}}) &\leq f(\boldsymbol{\theta}^k) + \nabla f(\boldsymbol{\theta}^k)^T(\boldsymbol{\theta}^{k+\frac{2}{3}} - \boldsymbol{\theta}^k) + \frac{L}{2}\|\boldsymbol{\theta}^{k+\frac{2}{3}} - \boldsymbol{\theta}^k\|^2 \\
&\leq f(\boldsymbol{\theta}^k) + (\boldsymbol{F}^{k^{-1}}\nabla f(\boldsymbol{\theta}^k))^T(\boldsymbol{\theta}^{k+\frac{2}{3}} - \boldsymbol{\theta}^k) + Q + \frac{L}{2}\|\boldsymbol{\theta}^{k+\frac{2}{3}} - \boldsymbol{\theta}^k\|^2 \\
&\leq f(\boldsymbol{\theta}^k) - \frac{1}{\eta^k}(\boldsymbol{\theta}^{k+\frac{2}{3}} - \boldsymbol{\theta}^k)^T(\boldsymbol{\theta}^{k+\frac{2}{3}} - \boldsymbol{\theta}^k) + Q + \frac{L}{2}\|\boldsymbol{\theta}^{k+\frac{2}{3}} - \boldsymbol{\theta}^k\|^2 \\
&= f(\boldsymbol{\theta}^k) - \frac{L}{2}\|\boldsymbol{\theta}^{k+\frac{2}{3}} - \boldsymbol{\theta}^k\|^2 - \nabla f(\boldsymbol{\theta}^{k+\frac{2}{3}})^T(\boldsymbol{\theta}^{k+1} - \boldsymbol{\theta}^{k+\frac{2}{3}}) - \frac{L}{2}\|\boldsymbol{\theta}^{k+1} - \boldsymbol{\theta}^{k+\frac{2}{3}}\|^2,
\end{aligned}
\tag{49}
$$

where $Q := \nabla f(\boldsymbol{\theta}^k)^T(\boldsymbol{\theta}^{k+\frac{2}{3}} - \boldsymbol{\theta}^k) - (\boldsymbol{F}^{k^{-1}}\nabla f(\boldsymbol{\theta}^k))^T(\boldsymbol{\theta}^{k+\frac{2}{3}} - \boldsymbol{\theta}^k)$, which represents the difference between the gradient and the nature gradient, and the equality follows by setting $\delta$ (*i.e.,* the size of the trust region) such that

$$
\eta^k = \frac{\|\boldsymbol{\theta}^{k+\frac{2}{3}} - \boldsymbol{\theta}^k\|^2}{L\|\boldsymbol{\theta}^{k+\frac{2}{3}} - \boldsymbol{\theta}^k\|^2 + Q + \nabla f(\boldsymbol{\theta}^{k+\frac{2}{3}})^T(\boldsymbol{\theta}^{k+1} - \boldsymbol{\theta}^{k+\frac{2}{3}}) + \frac{L}{2}\|\boldsymbol{\theta}^{k+1} - \boldsymbol{\theta}^{k+\frac{2}{3}}\|^2}.
$$

Again, based on Lemma D.2, for $\boldsymbol{\theta} \in \mathcal{C}_2$ we have

$$
\begin{aligned}
&(\boldsymbol{\theta}^k - \eta^k \boldsymbol{F}^{k^{-1}}\nabla f(\boldsymbol{\theta}^k) - \boldsymbol{\theta}^{k+\frac{2}{3}})(\boldsymbol{\theta} - \boldsymbol{\theta}^{k+\frac{2}{3}}) \leq 0 \\
\Rightarrow\quad &(-\eta^k \boldsymbol{F}^{k^{-1}}\nabla f(\boldsymbol{\theta}^k))^T(\boldsymbol{\theta} - \boldsymbol{\theta}^{k+\frac{2}{3}}) \leq -(\boldsymbol{\theta}^k - \boldsymbol{\theta}^{k+\frac{2}{3}})^T(\boldsymbol{\theta} - \boldsymbol{\theta}^{k+\frac{2}{3}}) \\
\Rightarrow\quad &\nabla f(\boldsymbol{\theta}^k)^T \boldsymbol{F}^{k^{-1}}(\boldsymbol{\theta} - \boldsymbol{\theta}^{k+\frac{2}{3}}) \geq \frac{1}{\eta^k}(\boldsymbol{\theta}^k - \boldsymbol{\theta}^{k+\frac{2}{3}})^T(\boldsymbol{\theta} - \boldsymbol{\theta}^{k+\frac{2}{3}}) \\
\Rightarrow\quad &\nabla f(\boldsymbol{\theta}^k)^T \boldsymbol{F}^{k^{-1}}\boldsymbol{\theta} \geq \nabla f(\boldsymbol{\theta}^k)^T \boldsymbol{F}^{k^{-1}}\boldsymbol{\theta}^{k+\frac{2}{3}} + \frac{1}{\eta^k}(\boldsymbol{\theta}^k - \boldsymbol{\theta}^{k+\frac{2}{3}})^T(\boldsymbol{\theta} - \boldsymbol{\theta}^{k+\frac{2}{3}}) \\
\Rightarrow\quad &\nabla f(\boldsymbol{\theta}^k)^T \boldsymbol{F}^{k^{-1}}(\boldsymbol{\theta} - \boldsymbol{\theta}^k) \geq \nabla f(\boldsymbol{\theta}^k)^T \boldsymbol{F}^{k^{-1}}(\boldsymbol{\theta}^{k+\frac{2}{3}} - \boldsymbol{\theta}^k) + \frac{1}{\eta^k}(\boldsymbol{\theta}^k - \boldsymbol{\theta}^{k+\frac{2}{3}})^T(\boldsymbol{\theta} - \boldsymbol{\theta}^{k+\frac{2}{3}}) \\
&\geq -\|\nabla f(\boldsymbol{\theta}^k)\|\|\boldsymbol{F}^{k^{-1}}\|\|\boldsymbol{\theta}^{k+\frac{2}{3}} - \boldsymbol{\theta}^k\| - \frac{1}{\eta^k}\|\boldsymbol{\theta}^{k+\frac{2}{3}} - \boldsymbol{\theta}^k\|\|\boldsymbol{\theta} - \boldsymbol{\theta}^{k+\frac{2}{3}}\| \\
&\geq -\big(G\sigma_1(\boldsymbol{F}^{k^{-1}}) + \frac{D}{\eta^k}\big)\|\boldsymbol{\theta}^{k+\frac{2}{3}} - \boldsymbol{\theta}^k\|,
\end{aligned}
\tag{50}
$$

where in the last two inequalities we use the property of the norm. Before reaching an $\epsilon$-FOSP, Eq. (50) implies that

$$
\begin{aligned}
-\epsilon &\geq \min_{\boldsymbol{\theta} \in \mathcal{C}_2} \nabla f(\boldsymbol{\theta}^k)^T \boldsymbol{F}^{k^{-1}}(\boldsymbol{\theta} - \boldsymbol{\theta}^k) \geq -\big(G\sigma_1(\boldsymbol{F}^{k^{-1}}) + \frac{D}{\eta^k}\big)\|\boldsymbol{\theta}^{k+\frac{2}{3}} - \boldsymbol{\theta}^k\| \\
\Rightarrow\quad &\|\boldsymbol{\theta}^{k+\frac{2}{3}} - \boldsymbol{\theta}^k\| \geq \frac{\epsilon}{\big(G\sigma_1(\boldsymbol{F}^{k^{-1}}) + \frac{D}{\eta^k}\big)}.
\end{aligned}
\tag{51}
$$

Based on Eq. (49) and Eq. (51), we have

$$
\begin{aligned}
f(\boldsymbol{\theta}^{k+\frac{2}{3}}) &\leq f(\boldsymbol{\theta}^k) - \frac{L}{2}\|\boldsymbol{\theta}^{k+\frac{2}{3}} - \boldsymbol{\theta}^k\|^2 - \nabla f(\boldsymbol{\theta}^{k+\frac{2}{3}})^T(\boldsymbol{\theta}^{k+1} - \boldsymbol{\theta}^{k+\frac{2}{3}}) - \frac{L}{2}\|\boldsymbol{\theta}^{k+1} - \boldsymbol{\theta}^{k+\frac{2}{3}}\|^2 \\
&\leq f(\boldsymbol{\theta}^k) - \frac{L\epsilon^2}{2(G\sigma_1(\boldsymbol{F}^{k^{-1}}) + \frac{D}{\eta^k})^2} - \nabla f(\boldsymbol{\theta}^{k+\frac{2}{3}})^T(\boldsymbol{\theta}^{k+1} - \boldsymbol{\theta}^{k+\frac{2}{3}}) - \frac{L}{2}\|\boldsymbol{\theta}^{k+1} - \boldsymbol{\theta}^{k+\frac{2}{3}}\|^2.
\end{aligned}
\tag{52}
$$

Based on the $L$-Lipschitz continuity of gradients, for the projection to the constraint set $\mathcal{C}_1$ we have

$$
f(\boldsymbol{\theta}^{k+1}) \leq f(\boldsymbol{\theta}^{k+\frac{2}{3}}) + \nabla f(\boldsymbol{\theta}^{k+\frac{2}{3}})^T(\boldsymbol{\theta}^{k+1} - \boldsymbol{\theta}^{k+\frac{2}{3}}) + \frac{L}{2}\|\boldsymbol{\theta}^{k+1} - \boldsymbol{\theta}^{k+\frac{2}{3}}\|^2.
\tag{53}
$$

Combining Eq. (52) with Eq. (53), we have

$$
f(\boldsymbol{\theta}^{k+1}) \leq f(\boldsymbol{\theta}^k) - \frac{L\epsilon^2}{2(G\sigma_1(\boldsymbol{F}^{k^{-1}}) + \frac{D}{\eta^k})^2}.
\tag{54}
$$

Hence it takes $\mathcal{O}(\epsilon^{-2})$ iterations to reach an $\epsilon$-FOSP. $\qquad\square$

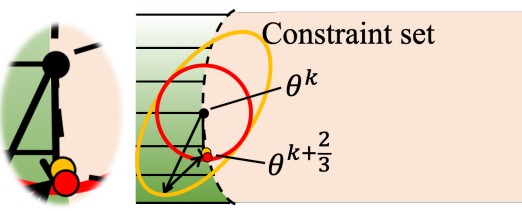

(a) SPACE under the KL-divergence projection

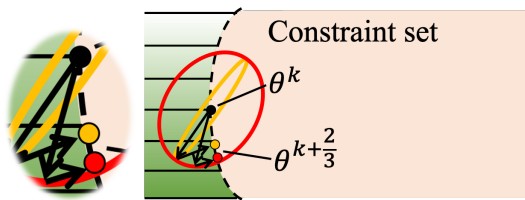

(b) SPACE under the 2-norm projection

Figure 6: Update procedures for SPACE under the KL and 2-norm projections with two possible Fisher information matrices. A lower objective value is achieved at the darker green area. Red and orange ellipses are $\boldsymbol{F}^k$s with two different spectra of singular values. Red and orange dots are resulting updated points under these two spectra of $\boldsymbol{F}^k$s. **(a)** A red ellipse has a smaller $\sigma_1(\boldsymbol{F}^k)$ and an orange ellipse has a larger $\sigma_1(\boldsymbol{F}^k)$. Both ellipses have the same $\sigma_n(\boldsymbol{F}^k)$. The two resulting $\boldsymbol{\theta}^{k+\frac{2}{3}}$ are similar. **(b)** A red ellipse has a larger $\sigma_n(\boldsymbol{F}^k)$ and an orange ellipse has a smaller $\sigma_n(\boldsymbol{F}^k)$. Both ellipses have the same $\sigma_1(\boldsymbol{F}^k)$. $\boldsymbol{\theta}^{k+\frac{2}{3}}$ with a larger $\sigma_n(\boldsymbol{F}^k)$ (red dot) has greater decrease of the objective value.

**Comments on Assumption 1.3.** In the paper, we assume that both the diameters of the cost constraint set ($\mathcal{C}_1$) and the region around $\pi_B$ ($\mathcal{C}_2$) are bounded above by $H$. This implies that given a small value for $h_D$, the convergence speed is determined by how large the constraint set is. This allows us to do an analysis for the algorithm. In practice, we agree that this assumption is too strong and leave it as a future work for improvement.

**Interpretation on Theorem 5.1.** We now provide a visualization in Fig. 6 under two possible projections. For each projection, we consider two possible Fisher information matrices. Please read the caption for more detail. In Fig. 6(a) we observe that since the reward improvement and projection steps use the KL-divergence, the resulting two update points with different $\sigma_1(\boldsymbol{F}^k)$ are similar. In addition, under the 2-norm projection, the larger $\sigma_n(\boldsymbol{F}^k)$ is, the greater the decrease in the objective. This is because that a large $\sigma_n(\boldsymbol{F}^k)$ implies a large curvature of $f$ in all directions. Intuitively, this makes the learning algorithm confident about where to update the policy to decrease the objective value greatly. Geometrically, a large $\sigma_n(\boldsymbol{F}^k)$ makes the 2-norm distance between the pre-projection and post-projection points small, leading to a small deviation from the reward improvement direction. This is illustrated in Fig. 6(b). We observe that since $\boldsymbol{F}^k$ determines the curvature of $f$ and the 2-norm projection is used, the updated point with a larger $\sigma_n(\boldsymbol{F}^k)$ (red dot) achieves more improvement of the objective value. These observations imply that the spectrum of the Fisher information matrix does not play a major role in SPACE under the KL-divergence projection, whereas it affects the decrease of objective value in SPACE under the 2-norm projection. Hence we choose either KL-divergence or 2-norm projections depending on the tasks to achieve better performance.

# E    ADDITIONAL EXPERIMENT RESULTS

## E.1    IMPLEMENTATION DETAILS

**Mujoco Task (Achiam et al., 2017).** In the point circle and ant circle tasks, the reward and cost functions are

$$R(s) = \frac{\boldsymbol{v}^T[-x_2; x_1]}{1 + |\,\|[x_1; x_2]\| - d\,|},$$

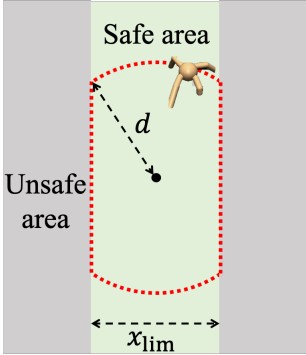

Figure 7: The environment of the circle task (adapted from Achiam et al. (2017)). The agent receives the maximum reward while staying in the safe area by following the red dashed line path.

and

$$C(s) = \mathbb{1}[|x_1| > x_{\text{lim}}],$$

where $x_1$ and $x_2$ are the coordinates in the plane, $\boldsymbol{v}$ is the velocity of the agent, and $d$, $x_{\text{lim}}$ are environmental parameters that specify the safe area. The agent is rewarded for moving fast in a wide circle with radius of $d$, but is constrained to stay within a safe region smaller than the radius of the circle in $x_1$-coordinate $x_{\text{lim}} \leq d$. For the point agent, we use $d = 5$ and $x_{\text{lim}} = 2.5$; for the ant agent, we use $d = 5$ and $x_{\text{lim}} = 1$. The environment is illustrated in Fig. 7.

In the point gather task, the agent receives a reward of $+10$ for gathering green apples, and a cost of 1 for gathering red apples. Two green apples and eight red apples are placed in the environment at the beginning. In the ant gather task, the agent receives a reward of $+10$ for gathering green apples, and a cost of 1 for gathering red apples. The agent also gets a reward of $-10$ for falling down to encourage smooth moving. Eight green apples and eight red apples are placed in the environment at the beginning.

For the point and ant agents, the state space consists of the positions, orientations, velocities, and the external forces applied to the torso and joint angles. The action space is the force applied to joints.

**Traffic Management Task (Vinitsky et al., 2018).** In the grid task, the state space, action space, reward function, and cost function are illustrated as follows.

(1) States: Speed, distance to the intersection, and edge number of each vehicle. The edges of the grid are uniquely numbered so the travel direction can be inferred. For the traffic lights, we return 0 and 1 corresponding to green or red for each light, a number between $[0, t_{\text{switch}}]$ indicating how long until a switch can occur, and 0 and 1 indicating if the light is currently yellow. Finally, we return the average density and velocity of each edge.

(2) Actions: A list of numbers $a = [-1, 1]^n$ where $n$ is the number of traffic lights. If $a_i > 0$ for traffic light $i$ it switches, otherwise no action is taken.

(3) Reward: The objective of the agent is to achieve high speeds. The reward function is

$$R(s) = \frac{\max(v_{\text{target}} - \|\boldsymbol{v}_{\text{target}} - \boldsymbol{v}\|, 0)}{v_{\text{target}}},$$

where $v_{\text{target}}$ is an arbitrary large velocity used to encourage high speeds and $\boldsymbol{v} \in \mathbb{R}^k$ is the velocities of $k$ vehicles in the network.

(4) Cost: The objective of the agent is to let lights stay red for at most 7 consecutive seconds. The cost function is

$$C(s) = \sum_{i=1}^{n} \mathbb{1}[t_{i,\text{red}} > 7],$$

where $t_{i,\text{red}}$ is the consecutive time that the light $i$ is in red.

In the bottleneck task, the state space, action space, reward function, and cost function are illustrated as follows.

(1) States: The states include: the mean positions and velocities of human drivers for each lane for each edge segment, the mean positions and velocities of the autonomous vehicles on each segment, and the outflow of the system in vehicles per/hour over the last 5 seconds.

(2) Actions: For a given edge-segment and a given lane, the action shifts the maximum speed of all the autonomous vehicles in the segment from their current value. By shifting the max-speed to higher or lower values, the system indirectly controls the velocity of the autonomous vehicles.

(3) Reward: The objective of the agent is to maximize the outflow of the whole traffic. The reward function is

$$R(s_t) = \sum_{i=t-\frac{5}{\Delta t}}^{i=t} \frac{n_{\text{exit}}(i)}{\frac{5}{\Delta t \cdot n_{\text{lane}} \cdot 500}},$$

where $n_{\text{exit}}(i)$ is the number of vehicles that exit the system at time-step $i$, and $n_{\text{lane}}$ is the number of lanes.

(4) Cost: The objective of the agent is to let the velocities of human drivers have lowspeed for no more than 10 seconds. The cost function is

$$C(s) = \sum_{i=1}^{n_{\text{human}}} \mathbb{1}[t_{i,\text{low}} > 10],$$

where $n_{\text{human}}$ is the number of human drivers, and $t_{i,\text{low}}$ is the consecutive time that the velocity of human driver $i$ is less than 5 m/s. For more information, please refer to Vinitsky et al. (2018).

**Car-racing Task.** In the car-racing task, the state space, action space, reward function, and the cost function are illustrated as follows.

(1) States: It is a high-dimensional space where the state is a $96 \times 96 \times 3$ tensor of raw pixels. Each pixel is in the range of $[0, 255]$.

(2) Actions: The agent has 12 actions in total: $a \in \mathcal{A} = \{(a^{\text{steer}}, a^{\text{gas}}, a^{\text{brake}}) | a^{\text{steer}} \in \{-1, 0, 1\}, a^{\text{gas}} \in \{0, 1\}, a^{\text{brake}} \in \{0, 0.2\}\}$, where $a^{\text{steer}}$ is the steering angle, $a^{\text{gas}}$ is the amount of gas applied, and $a^{\text{brake}}$ is the amount of brake applied.

(3) Reward: In each episode, we randomly generate the track. The episode is terminated if the agent reaches the maximal step or traverse over 95% of the track. The track is discretized into 281 tiles. The agent receives a reward of $\frac{1000}{281}$ for each tile visited. To encourage driving efficiency, the agent receives a penalty of $-1$ per-time step.

(4) Cost: The cost is to constrain the accumulated number of brakes to encourage a smooth ride.

**Architectures and Parameters.** For the gather and circle tasks we test two distinct agents: a point-mass ($S \subseteq \mathbb{R}^9, A \subseteq \mathbb{R}^2$), and an ant robot ($S \subseteq \mathbb{R}^{32}, A \subseteq \mathbb{R}^8$). The agent in the grid task is $S \subseteq \mathbb{R}^{156}, A \subseteq \mathbb{R}^4$, and the agent in the bottleneck task is $S \subseteq \mathbb{R}^{141}, A \subseteq \mathbb{R}^{20}$. Finally, the agent in the car-racing task is $S \subseteq \mathbb{R}^{96 \times 96 \times 3}, A \subseteq \mathbb{R}^3$.

For the simulations in the gather and circle tasks, we use a neural network with two hidden layers of size (64, 32) to represent Gaussian policies. And we use the KL-divergence projection. For the simulations in the grid and bottleneck tasks, we use a neural network with two hidden layers of size (16, 16) and (50, 25) to represent Gaussian policies, respectively. And we use the 2-norm projection. For the simulation in the car-racing task, we use a convolutional neural network with two convolutional operators of size 24 and 12 followed by a dense layer of size (32, 16) to represent a Gaussian policy. And we use the KL-divergence projection. The choice of the projections depends on the task itself, we report the best performance among two projections. We use $\tanh$ as an activation function for all the neural network policies. In the experiments, since the step size is small, we reuse the Fisher information matrix of the reward improvement step in the KL-divergence projection step to reduce the computational cost.

| Parameter | PC | PG | AC | AG | Gr | BN | CR |
|---|---|---|---|---|---|---|---|
| Reward dis. factor $\gamma$ | 0.995 | 0.995 | 0.995 | 0.995 | 0.999 | 0.999 | 0.990 |
| Constraint cost dis. factor $\gamma_C$ | 1.0 | 1.0 | 1.0 | 1.0 | 1.0 | 1.0 | 1.0 |
| Divergence cost dis. factor $\gamma_D$ | 1.0 | 1.0 | 1.0 | 1.0 | 1.0 | 1.0 | 1.0 |
| step size $\delta$ | $10^{-4}$ | $10^{-4}$ | $10^{-4}$ | $10^{-4}$ | $10^{-4}$ | $10^{-4}$ | $5 \times 10^{-4}$ |
| $\lambda_R^{\text{GAE}}$ | 0.95 | 0.95 | 0.95 | 0.95 | 0.97 | 0.97 | 0.95 |
| $\lambda_C^{\text{GAE}}$ | 1.0 | 1.0 | 0.5 | 0.5 | 0.5 | 1.0 | 1.0 |
| $\lambda_D^{\text{GAE}}$ | 0.95 | 0.95 | 0.95 | 0.95 | 0.90 | 0.90 | 0.95 |
| Batch size | 50,000 | 50,000 | 100,000 | 100,000 | 10,000 | 25,000 | 10,000 |
| Rollout length | 50 | 15 | 500 | 500 | 400 | 500 | 1000 |
| Constraint cost threshold $h_C$ | 5 | 0.5 | 5 | 0.2 | 0 | 0 | 5 |
| Divergence cost threshold $h_D^0$ | 5 | 3 | 5 | 3 | 10 | 10 | 5 |
| Number of policy updates | 1,000 | 1,200 | 2,500 | 1,500 | 200 | 300 | 600 |

Table 2: Parameters used in all tasks. (PC: point circle, PG: point gather, AC: ant circle, AG: ant gather, Gr: grid, BN: bottleneck, and CR: car-racing tasks)

|  | PCPO | | SPACE (Ours) | | f-PCPO | | f-CPO | | d-PCPO | | d-CPO | |
|---|---|---|---|---|---|---|---|---|---|---|---|---|
|  | M/C | Time | M/C | Time | M/C | Time | M/C | Time | M/C | Time | M/C | Time |
| PG | B | 22.14 | B | 25.2 | B | 31.9 | B | 25.5 | B | 32.8 | B | 32.6 |
| PC | B | 35.1 | B | 51.2 | B | 48.4 | B | 49.4 | B | 55.5 | B | 55.9 |
| AG | B | 386.9 | B | 110.5 | C | 268.6 | C | 235.1 | B | 138.2 | B | 187.5 |
| AC | B | 148.9 | B | 94.0 | C | 222.6 | C | 214.6 | B | 177.4 | B | 151.2 |
| Gr | A | 105.3 | A | 91.4 | A | 88.2 | A | 58.7 | A | 116.8 | A | 115.3 |
| BN | A | 257.7 | A | 181.1 | A | 162.9 | A | 161.6 | A | 259.3 | A | 275.6 |
| CR | C | 993.5 | C | 971.6 | C | 1078.3 | C | 940.1 | C | 1000.4 | C | 981.0 |

Table 3: Real-time in seconds for one policy update for all tested algorithms and tasks. (PC: point circle, PG: point gather, AC: ant circle, AG: ant gather, Gr: grid, BN: bottleneck, and CR: car-racing tasks)

We use GAE-$\lambda$ approach (Schulman et al., 2016) to estimate $A_R^\pi(s, a)$, $A_C^\pi(s, a)$, and $A_D^\pi(s)$. For the simulations in the gather, circle, and car-racing tasks, we use neural network baselines with the same architecture and activation functions as the policy networks. For the simulations in the grid and bottleneck tasks, we use linear baselines. The hyperparameters of all algorithms and all tasks are in Table 2.

We conduct the experiments on three separate machines: machine A has an Intel Core i7-4770HQ CPU, machine B has an Intel Core i7-6850K CPU, and machine C has an Intel Xeon X5675 CPU. We report real-time (*i.e.,* wall-clock time) in seconds for one policy update for all tested algorithms and tasks in Table 3. We observe that SPACE has the same computational time as the other baselines.

For the most intensive task, *i.e.,* the car-racing task, the memory usage is 6.28GB. The experiments are implemented in rllab (Duan et al., 2016), a tool for developing RL algorithms. We provide the link to the code: `https://sites.google.com/view/spacealgo`.

**Comments on the rationale behind when to increase $h_D$.** The update method of $h_D$ is empirically designed to ensure that the value of the cost does not increase (*i.e.,* $J_C(\pi^k) \leq J_C(\pi^{k-1})$) and the reward keeps improving (*i.e.,* $J_R(\pi^k) \geq J_R(\pi^{k-1})$) after learning from $\pi_B$. Lemma 4.1 theoretically ensures $h_D$ is large enough to guarantee feasibility and exploration of the agent.

**Implementation of Updating $h_D^k$.** Lemma 4.1 shows that $h_D^{k+1}$ should be increased at least by $\mathcal{O}\big((J_C(\pi^k) - h_C)^2\big) + h_D^k$ if $J_C(\pi^k) > J_C(\pi^{k-1})$ or $J_R(\pi^k) < J_R(\pi^{k-1})$ at step $k$. We now provide the practical implementation. For each policy update we check the above conditions. If one of the conditions satisfies, we increase $h_D^{k+1}$ by setting the constant to 10, *i.e.,* $10 \cdot (J_C(\pi^k) - h_C)^2 + h_D^k$. In practice, we find that the performance of SPACE is not affected by the selection of the constant. Note that we could still compute the exact value of $h_D^{k+1}$ as shown in the proof of Lemma 4.1. However, this incurs the computational cost.

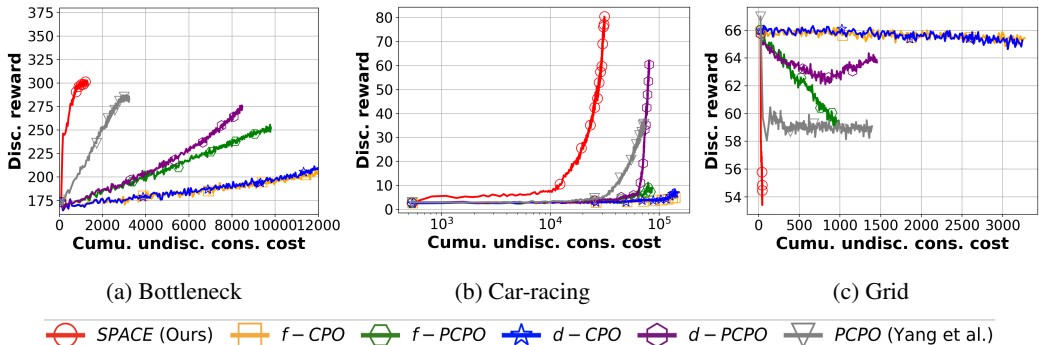

Figure 8: The discounted reward vs. the cumulative undiscounted constraint cost over policy updates for the tested algorithms and tasks. The solid line is the mean over 5 runs. SPACE achieves the same reward performance with fewer cost constraint violations in all cases. (Best viewed in color.)

**Comments on learning from multiple baseline policies $\pi_B$.** In our setting, we use one $\pi_B$. This allows us to do theoretical analysis. One possible idea for learning from multiple $\pi_B$ is to compute the distance to each $\pi_B$. Then, select the one with the minimum distance to do the update. This ensures that the update for the reward in the first step is less affected by $\pi_B$. And the analysis we did can be extended. We leave it as future work for developing this.

**Comments on refining the PCPO agent's policy (Yang et al., 2020) directly.** Fine-tuning the pre-trained policy directly might result in lower reward and cost violations. This is because that the pre-trained policy has a low entropy and it does not explore. We empirically observe that the agent pre-trained with the baseline policy yields less reward in the new task (*i.e.,* different cost constraint thresholds $h_C$) as illustrated in Section E.2. In contrast, the SPACE agent simultaneously learns from the baseline policy while ensuring the policy entropy is high enough to explore the environment.

**Comments on the feasibility of getting safe baseline policies.** In many real-world applications such as drones, we can obtain baseline policies modeled from the first principle physics, or pre-train baseline policies in the constrained and safe environment, or use rule-based baseline policies. Importantly, we do not assume the baseline has to be a "safe policy" – it can be a heuristic that ignores safety constraints. This is one of the main motivations for our algorithm: to utilize priors from the baseline which may be unsafe, but guarantee the safety of the newly learned algorithm according to the provided constraints.

**Instructions for Reproducibility.** We now provide the instructions for reproducing the results. First install the libraries for python3 such as numpy, scipy. To run the Mujoco experiments, get the licence from `https://www.roboti.us/license.html`. To run the traffic management experiments, install FLOW simulator from `https://flow.readthedocs.io/en/latest/`. To run the car-racing experiments, install OpenAI Gym from `https://github.com/openai/gym`. Our implementation is based on the environment from Achiam et al. (2017), please download the code from `https://github.com/jachiam/cpo`. The code is based on rllab (Duan et al., 2016), install the relevant packages such as theano (`http://deeplearning.net/software/theano/`). Then, download SPACE code from `https://sites.google.com/view/spaceneurips` and place the codes on the designated folder instructed by Readme.txt on the main folder. Finally, go to the example folder and execute the code using python command.

## E.2 EXPERIMENT RESULTS

**Baseline policies.** We pre-train the baseline policies using a safe RL algorithm. Here we also consider three types of baseline policies: **(1)** *suboptimal* $\pi_B^{\text{cost}}$ with $J_C(\pi_B^{\text{cost}}) \approx 0$, **(2)** *suboptimal* $\pi_B^{\text{reward}}$ with $J_C(\pi_B^{\text{reward}}) > h_C$, and **(3)** $\pi_B^{\text{near}}$ with $J_C(\pi_B^{\text{near}}) \approx h_C$ Note that these $\pi_B$ have different degrees of constraint satisfaction.

**The Discounted Reward vs. the Cumulative Undiscounted Constraint Cost (see Fig. 8).** To show that SPACE achieves higher reward under the same cost constraint violations (*i.e.,* learning a

constraint-satisfying policy without violating the cost constraint a lot), we examine the discounted reward versus the *cumulative* undiscounted constraint cost. The learning curves of the discounted reward versus the cumulative undiscounted constraint cost are shown for all tested algorithms and tasks in Fig. 8. We observe that in these tasks under the same value of the reward, SPACE outperforms the baselines significantly with fewer cost constraint violations. For example, in the car-racing task SPACE achieves 3 times fewer cost constraint violations at the reward value of 40 compared to the best baseline – PCPO. This implies that SPACE effectively leverages the baseline policy while ensuring the constraint satisfaction. In contrast, without the supervision of the baseline policy, PCPO requires much more constraint violations to achieve the same reward performance as SPACE. In addition, although the fixed-point and the dynamic-point approaches use the supervision of the baseline policy, the lack of the projection step makes them less efficient in learning a constraint-satisfying policy.

**Comparison of Baseline Policies (see Fig. 9).** To examine whether SPACE can safely learn from the baseline policy which need not satisfy the cost constraint, we consider two baseline policies: $\pi_B^{\text{cost}}$ and $\pi_B^{\text{reward}}$. The learning curves of the undiscounted constraint cost, the discounted reward, and the undiscounted divergence cost with two possible baselines over policy updates are shown for all tested algorithms and tasks in Fig. 9. We observe that in the point gather and point circle tasks, the initial values of the cost are larger than $h_C$ (*i.e.*, $J_C(\pi^0) > h_C$). Using $\pi_B^{\text{cost}}$ allows the learning algorithm to quickly satisfy the cost without doing the extensive projection onto the cost constraint set. For example, in the point circle task we observe that learning guided by $\pi_B^{\text{cost}}$ quickly satisfies the cost constraint. In addition, we observe that in the ant gather and ant circle tasks, the initial values of the cost are smaller than $h_C$ (*i.e.*, $J_C(\pi^0) < h_C$). Intuitively, we would expect that using $\pi_B^{\text{reward}}$ allows the agent to quickly improve the reward since the agent already satisfies the cost constraint in the beginning. In the ant gather task we observe that SPACE guided by $\pi_B^{\text{reward}}$ does improve the reward more quickly at around 200 iteration. However, we observe that the agent guided by the both baseline policies achieve the same final reward performance in the ant gather and ant circle tasks. The reason is that using dynamic $h_D$ allows the agent to stay away from the baseline policy. This makes the baseline policy less influential in the end. As a result, the reward improvement mostly comes from the reward improvement step of SPACE if the agent starts in the interior of the cost constraint set (*i.e.*, $J_C(\pi^0) \leq h_C$).

**Fixed $h_D$ (see Fig. 10).** To understand the effect of using dynamic $h_D^k$ when learning from a sub-optimal baseline policy, we compare the performance of SPACE with and without adjusting $h_D$. The learning curves of the undiscounted constraint cost, the discounted reward, and the undiscounted divergence cost over policy updates are shown for all tested algorithms and tasks in Fig. 10. We observe that SPACE with fixed $h_D$ converges to less reward. For example, in the ant circle task SPACE with the dynamic $h_D$ achieves 2.3 times more reward. The value of the divergence cost in the ant circle task shows that staying away from the baseline policy achieves more reward. This implies that the baseline policy in the ant circle task is highly sub-optimal to the agent. In addition, we observe that in some tasks the dynamic $h_D$ does not have much effect on the reward performance. For example, in the point gather task SPACE achieves the same reward performance. The values of the divergence cost in the point gather task decrease throughout the training. These observations imply that the update scheme of $h_D$ is critical for some tasks.

**Comparison of SPACE vs. d-CPO, d-PCPO and the Pre-training Approach (see Fig. 11).** To show that SPACE is effective in using the supervision of the baseline policy, we compare the performance of SPACE to the dynamic-point and the pre-training approaches. In the pre-training approach, the agent first performs the trust region update with the objective function being the divergence cost. Once the agent has the same reward performance as the baseline policy (*i.e.*, $J_R(\pi^{\tilde{k}}) \approx J_R(\pi_B)$ for some $k$), the agent performs the trust region update with the objective function being the reward function. The learning curves of the undiscounted constraint cost, the discounted reward, and the undiscounted divergence cost over policy updates are shown for all tested algorithms and tasks in Fig. 11. We observe that SPACE achieves better reward performance compared to the pre-training approach in all tasks. For example, in the point circle, ant gather and ant circle tasks the pre-training approach seldom improves the reward but all satisfies the cost constraint. This implies that the baseline policies in these tasks are highly sub-optimal in terms of reward performance. In contrast, SPACE prevents the agent from converging to a poor policy.

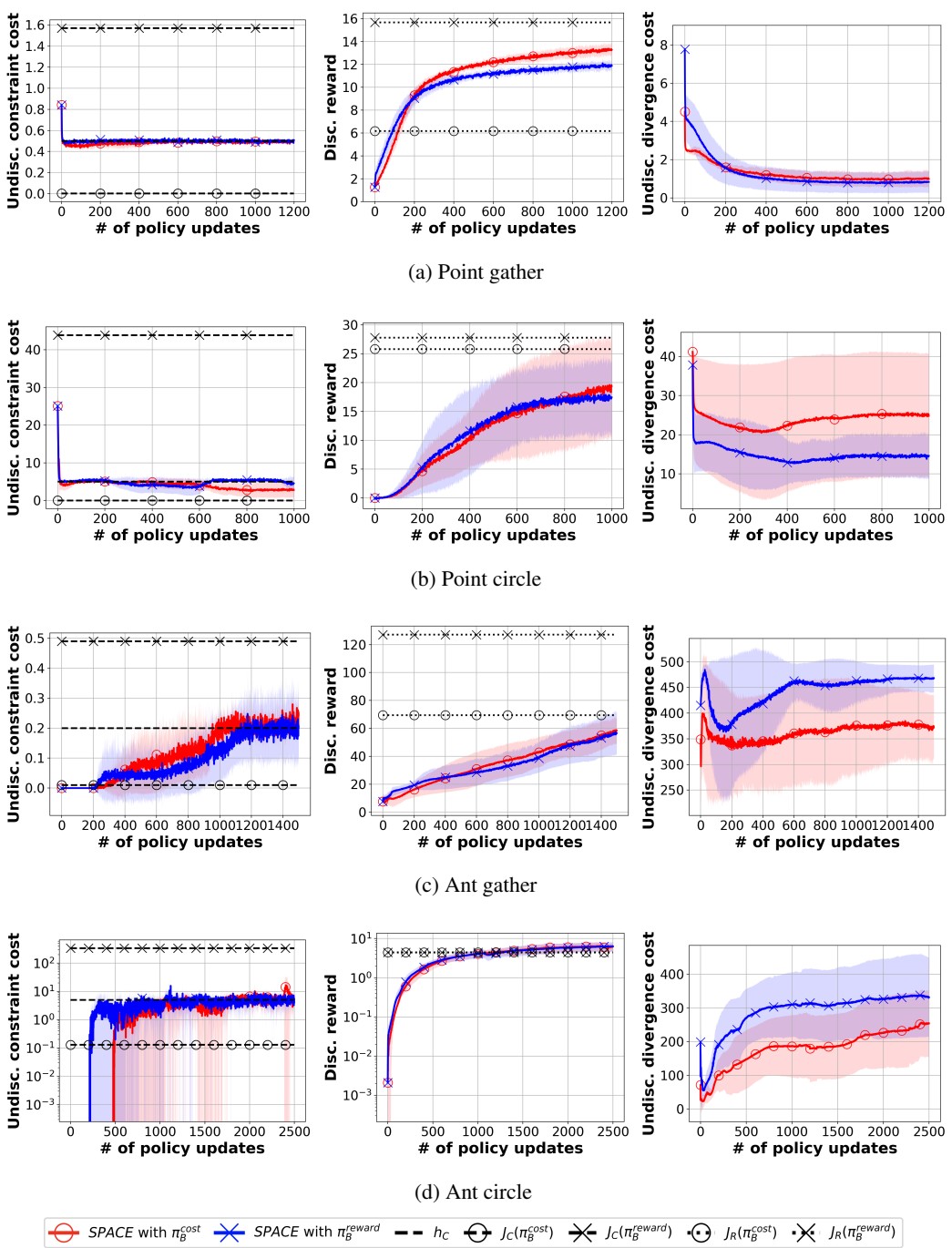

Figure 9: The undiscounted constraint cost, the discounted reward, and the undiscounted divergence cost over policy updates for the tested algorithms and tasks. The solid line is the mean and the shaded area is the standard deviation over 5 runs. SPACE ensures cost constraint satisfaction guided by the baseline policy which need not satisfy the cost constraint. (Best viewed in color.)

In addition, we observe that in the point gather task the pre-training approach achieves the same reward performance as the baseline policy, whereas SPACE has a better reward performance compared to the baseline policy. The pre-training approach does not keep improving the reward after learning from the baseline policy. This is because that after pre-training with the baseline policy, the entropy of the learned policy is small. This prevents the agent from trying new actions which may lead to better reward performance. This implies that pre-training approach may hinder the exploration of the learning agent on the new environment. Furthermore, in the car-racing task we observe that using pre-training approach achieves the same reward performance as SPACE but improves reward slowly,

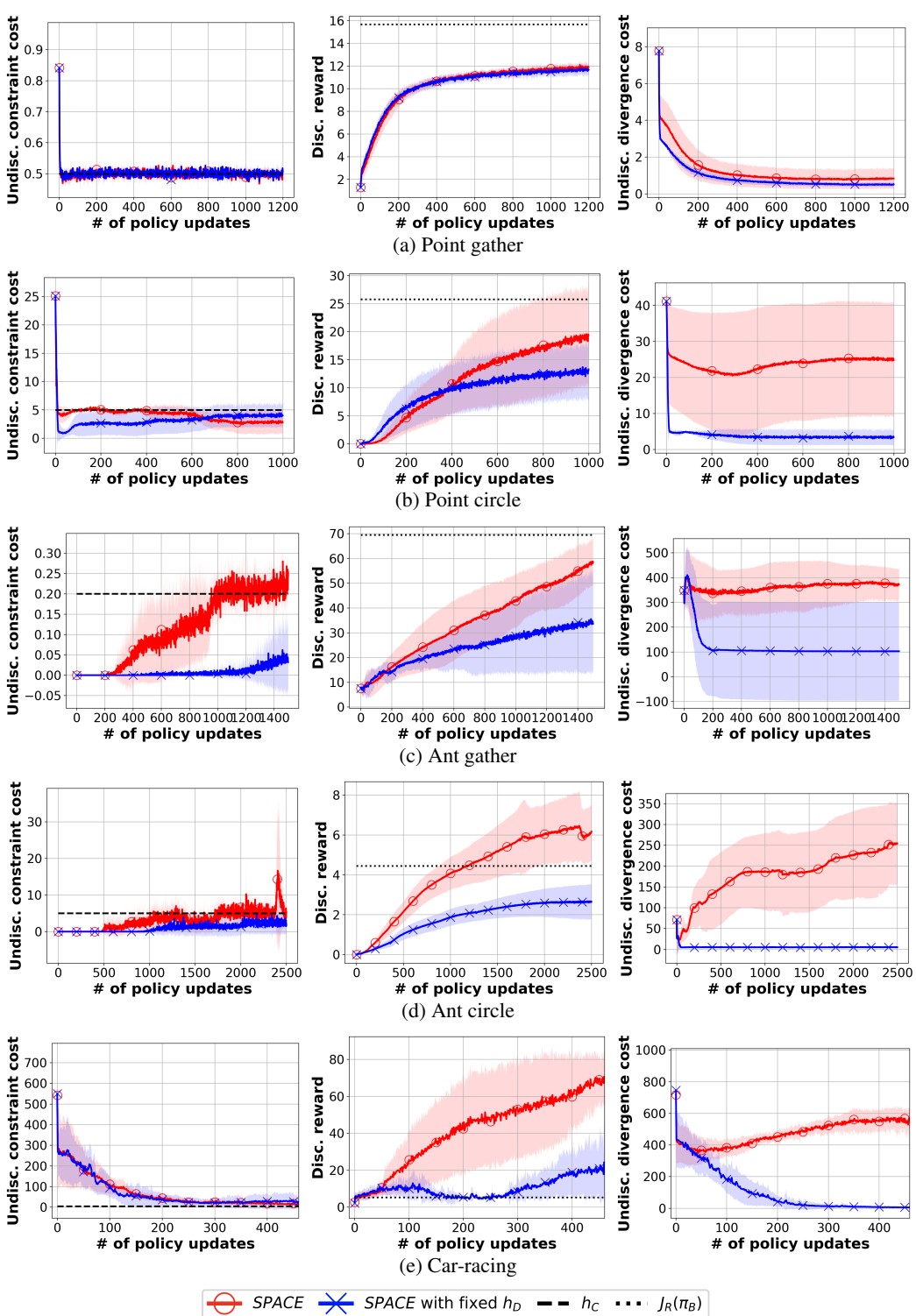

Figure 10: The undiscounted constraint cost, the discounted reward, and the undiscounted divergence cost over policy updates for the tested algorithms and tasks. The solid line is the mean and the shaded area is the standard deviation over 5 runs. SPACE with the dynamic $h_D$ achieves higher reward. (Best viewed in color.)

and the pre-training approach has more cost constraint violations than SPACE. This implies that jointly using reinforcement learning and the supervision of the baseline policy achieve better reward and cost performance.

For d-CPO and d-PCPO, in the point and ant tasks we observe that both approaches have comparable or silently better reward and cost performance compared to SPACE. However, in the car-racing task we observe that d-CPO cannot improve the reward due to a slow update procedure for satisfying the cost constraint, whereas d-PCPO has a better reward performance. These observations imply that the projection steps in SPACE allow the learning agent to effectively and robustly learn from the baseline policy.

**Comparison of SPACE under the KL-divergence and the 2-norm Projections (see Fig. 12).** Theorem 5.1 shows that under the KL-divergence and 2-norm projections, SPACE converges to different stationary points. To demonstrate the difference between these two projections, Fig. 12 shows the learning curves of the undiscounted constraint cost, the discounted reward, and the undiscounted divergence cost over policy updates for all tested algorithms and tasks. In the Mujoco tasks, we observe that SPACE under the KL-divergence projection achieves higher reward. For instance, in the point gather task the final reward is 25% higher under the same cost constraint satisfaction. In contrast, in the traffic management tasks, we observe that SPACE under the 2-norm projection achieves better cost constraint satisfaction. For instance, in the grid task SPACE under the 2-norm projection achieves a lower reward but more cost constraint satisfaction. In addition, in the bottleneck task SPACE under the 2-norm projection achieves more reward and cost constraint satisfaction. These observations imply that SPACE converges to different stationary points under two possible projections depending on tasks.

**Initial $h_D^0$ (see Fig. 13).** To understand the effect of the initial value of $h_D^0$, we test SPACE with three different initial values: $h_D^0 = 1, h_D^0 = 5$, and $h_D^0 = 25$ in the ant circle and car-racing tasks. The learning curves of the undiscounted constraint cost, the discounted reward, and the undiscounted divergence cost over policy updates are shown for all tested algorithms and tasks in Fig. 13. In both tasks, we observe that the initial value of $h_D^0$ does not affect the reward and the cost performance significantly (*i.e.,* the mean of learning curves lies in roughly the same standard deviation over the initialization). In addition, the value of the divergence cost over three $h_D^0$ are similar throughout the training. These observations imply that the update scheme of $h_D^k$ in SPACE is robust to the choice of the initial value of $h_D^0$.

However, in the car-racing task we observe that the learning curves of using a smaller $h_D^0$ tend to have higher variances. For example, the standard deviation of $h_D^0 = 1$ in the reward plot is 6 times larger than the one with $h_D^0 = 25$. This implies that SPACE may have reward performance degradation when using a smaller initial value of $h_D^0$. One possible reason is that when the distance between the learned and baseline policies is large, using a small value of $h_D^0$ results in an inaccurate projection (*i.e.,* due to approximation errors). This causes the policy to follow a zigzag path. We leave the improvement of this in future work.

## F  HUMAN POLICIES

We now describe the procedure for collecting human demonstration data in the car-racing task. A player uses the right key, left key, up key and down key to control the direction, acceleration, and brake of the car. The human demonstration data contain the display of the game (*i.e.,* the observed state), the actions, and the reward. We collect 20 minutes of demonstration data. A human player is instructed to stay in the lane but does not know the cost constraint. This allows us to test whether SPACE can safely learn from the baseline policy which need not satisfy the cost constraints. We then use an off-policy algorithm (DDPG) trained on the demonstration data to get the baseline human policy. Since the learned baseline human policy does not interact with the environment, its reward performance cannot be better than the human performance. Fig. 14 shows the procedure.

**Implementation Details of DDPG.** We use DDPG as our off-policy algorithm. We use a convolutional neural network with two convolutional operators of size 24 and 12 followed by a dense layer of size (32, 16) to represent a Gaussian policy. A Q function shares the same architecture of the policy. The learning rates of the policy and Q function are set to $10^{-4}$ and $10^{-3}$, respectively.

# G  THE MACHINE LEARNING REPRODUCIBILITY CHECKLIST (VERSION 1.2, MAR.27 2019)

For all models and algorithms presented, indicate if you include[2]:

- A clear description of the mathematical setting, algorithm, and/or model:
  - **Yes**, please see the problem formulation in Section 3, the update procedure for SPACE in Section 5, and the architecture of the policy in Section E.1.
- An analysis of the complexity (time, space, sample size) of any algorithm:
  - **Yes**, please see the implementation details in Section E.1.
- A link to a downloadable source code, with specification of all dependencies, including external libraries:
  - **Yes**, please see the implementation details in Section E.1.

For any theoretical claim, check if you include:

- A statement of the result:
  - **Yes**, please see Section 4 and Section 5.
- A clear explanation of any assumptions:
  - **Yes**, please see Section 4 and Section 5.
- A complete proof of the claim:
  - **Yes**, please see Section B, Section C, and Section D.

For all figures and tables that present empirical results, indicate if you include:

- A complete description of the data collection process, including sample size:
  - **Yes**, please see Section E.1 for the implementation details.
- A link to a downloadable version of the dataset or simulation environment:
  - **Yes**, please see Section E.1 for the simulation environment.
- An explanation of any data that were excluded, description of any pre-processing step:
  - **It's not applicable.** This is because that data comes from simulated environments.
- An explanation of how samples were allocated for training / validation / testing:
  - **It's not applicable.** The complete trajectories (*i.e.,* data) is used for training. There is no validation set. Testing is performed in the form of online learning approaches.
- The range of hyper-parameters considered, method to select the best hyper-parameter configuration, and specification of all hyper-parameters used to generate results:
  - **Yes**, we randomly select five random seeds, and please see Section E.1 for the implementation details.
- The exact number of evaluation runs:
  - **Yes**, please see Section E.1 for the implementation details.
- A description of how experiments were run:
  - **Yes**, please see Section E.1 for the implementation details.
- A clear definition of the specific measure or statistics used to report results:
  - **Yes**, please see Section 6.
- Clearly defined error bars:
  - **Yes**, please see Section 6.

---

[2]Here is a link to the list: `https://www.cs.mcgill.ca/~jpineau/ReproducibilityChecklist.pdf`.

- A description of results with central tendency (*e.g.,* mean) variation (*e.g.,* stddev):
    - **Yes**, please see Section 6.
- A description of the computing infrastructure used:
    - **Yes**, please see Section E.1 for the implementation details.

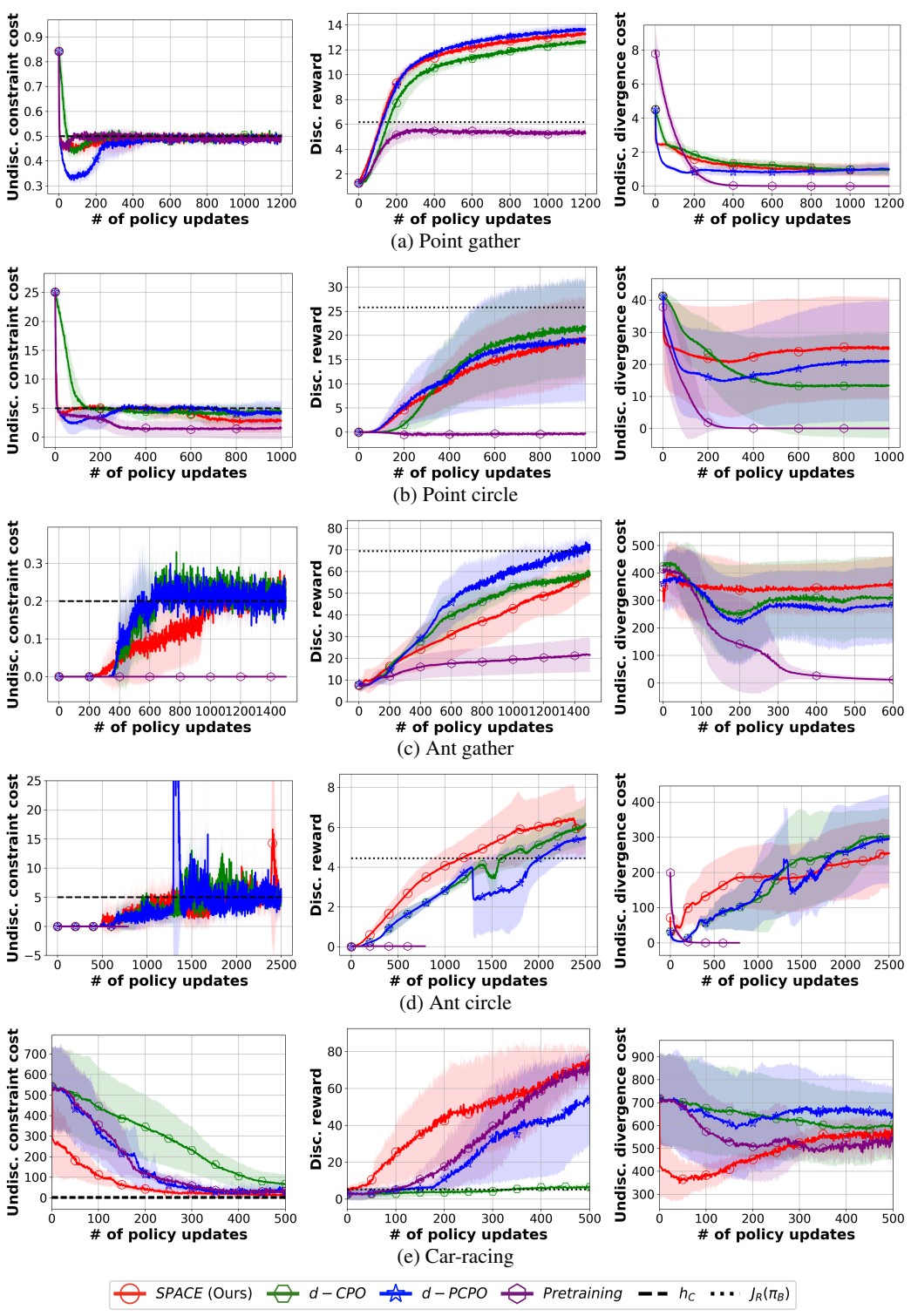

Figure 11: The undiscounted constraint cost, the discounted reward, and the undiscounted divergence cost over policy updates for the tested algorithms and tasks. The solid line is the mean and the shaded area is the standard deviation over 5 runs. SPACE outperforms d-CPO, d-PCPO and the pre-training approach in terms of the efficiency of the reward improvement and cost constraint satisfaction. (Best viewed in color.)

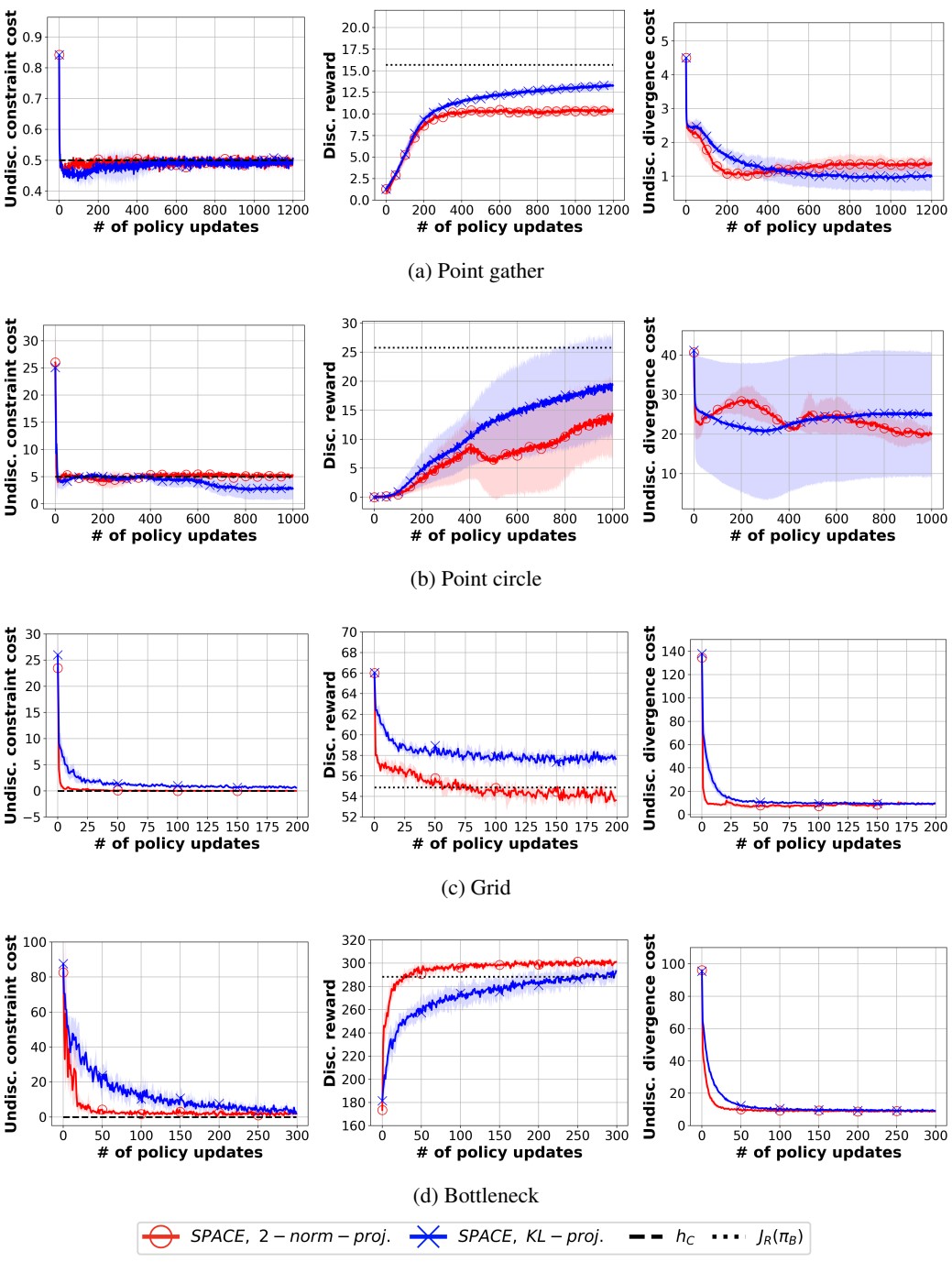

Figure 12: The undiscounted constraint cost, the discounted reward, and the undiscounted divergence cost over policy updates for the tested algorithms and tasks. The solid line is the mean and the shaded area is the standard deviation over 5 runs. SPACE converges to differently stationary points under two possible projections. (Best viewed in color.)

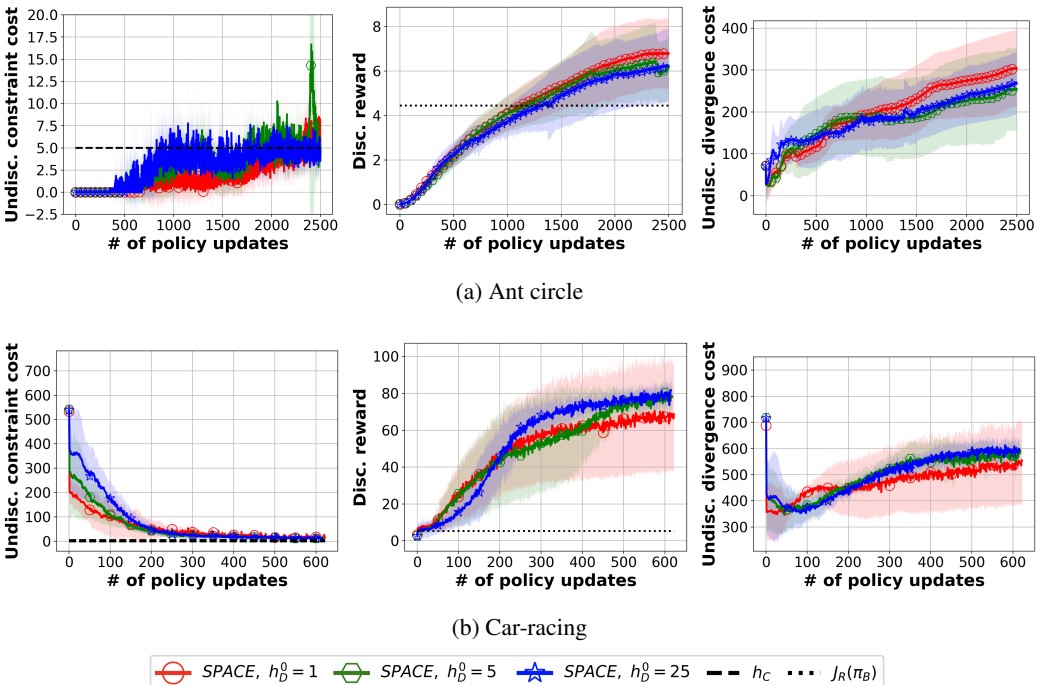

(a) Ant circle

(b) Car-racing

Figure 13: The undiscounted constraint cost, the discounted reward, and the undiscounted divergence cost over policy updates for the tested algorithms and tasks. The solid line is the mean and the shaded area is the standard deviation over 5 runs. We observe that the initial value of $h_D^0$ does not affect the reward and the cost performance significantly. (Best viewed in color.)

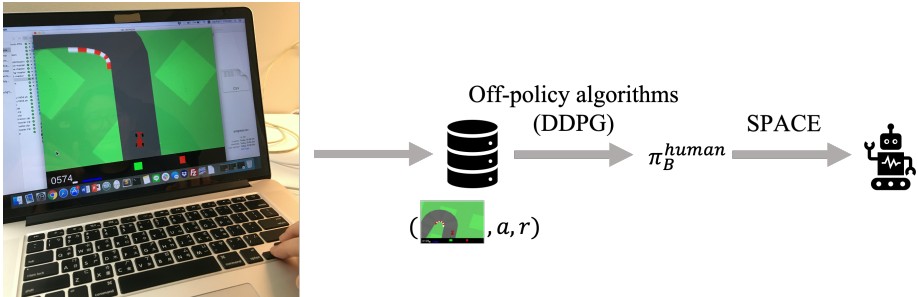

Figure 14: Procedure for getting a baseline human policy. We ask a human to play the car-racing game. He/She does not know the cost constraint. The trajectories (*i.e.,* display of the game, the action, and the reward) are then stored. A human policy is obtained by using an off-policy algorithm (DDPG) trained on the trajectories.

