# OpenReview forum: "Accelerating Safe Reinforcement Learning with Constraint-mismatched Policies"
_ICLR.cc/2021/Conference — Reject_

### Official Review · AnonReviewer4 · 2020-10-23
**Interesting work, but it might lack novelty**

**Rating:** 5
**Confidence:** 3

**Review:**

Summary

The authors propose SPACE, an RL algorithm for learning a policy that maximizes reward while satisfying given constraints in a setting where a baseline policy is provided. They design a three-step update rule for learning such a policy and provide a finite-sample analysis of the resulting method in a simplified setting. They report numerical results in several domains that show how the proposed approach outperforms competitive baselines.

Pros

- The paper is very well-written and easy to follow.
- The experimental evaluation is performed in complex domains. The results are significant and well-described. Full details are provided in the appendix to enable reproducibility.

Cons

- The paper might lack some novelty. The approach seems quite incremental with respect to Yang et al. 2020 (though I am not very familiar with that work to precisely judge this). In particular, handling the additional constraint due to the baseline policy seems not that different from how the "cost constraint" is handled, and indeed the two approaches have a very similar update rule. The experimental domains seem also the same (except for the car racing one which was not present in Yang et al. 2020).
- The setting considered for the finite-sample analysis might be overly-simplified. In particular, the analysis seems not really "finite-sample" but rather "finite-time", in the sense that it is carried out for a finite number of learning steps in a standard optimization setting where all functions (e.g., the gradients) can be exactly evaluated and are not estimated from finite samples.

Detailed comments

1. While it is intuitive why we need to project the policy onto the set of policies satisfying the constraint in Eq. 2, it was not very clear to me why we need to perform a projection to stay close to the baseline policy (especially since the actual "closeness" h_D is changed by the algorithm). It seems that, after all, what we care about is to stay close to this policy during the initial phases of learning (if possible) and slowly stop caring about this as learning proceeds. So could we simply regularize the standard RL objective with the distance wrt the baseline policy and make the regularization parameter fade over time? Would that perform worse than projecting?

2. Lemma 4.1 provides a formal justification for increasing h_D over time. How is the actual increase performed in practice? Is there any fixed "increase value" or is the theoretical value from the lemma used?

3. Could you provide some intuition on why the finite-sample bound for SPACE with KL divergence scales with the maximum eigenvalue of F^k while the one for l2-norm scales with the minimum eigenvalue of the same matrix?

4. It seems that the work of Yang et al. 2020 also reports a similar convergence guarantee as the one derived here. Could the authors clarify what are the technical difficulties in extending such a result to this setting? It seems to me that after the simplifications the two considered settings are quite similar.

5. In the experiments, the authors mention that "PCPO agent cannot robustly satisfy the cost constraints". Why does this happen? Doesn't PCPO handle the constraints in the same way as SPACE?

6. Why is there no baseline for comparison in Figure 4?

Some minor comments:

- In the caption of Table 1, I would say "w/ expert/optimal demonstrations" to be more precise and "constraint-cost satisfaction" (cost alone might be interpreted as the negative reward)
- Third line of page 8: "gird" -> "grid"

Overall comment

Overall I believe that the paper is interesting and it reports good results. However, at the present time, I am not sure about the novelty wrt Yang et al. 2020, so I am quite borderline. I will be happy to increase my score after the authors have clarified my doubts.

References

Tsung-Yen Yang, Justinian Rosca, Karthik Narasimhan, and Peter J. Ramadge. Projection-based
constrained policy optimization. In Proceedings of the International Conference on Learning
Representations, 2020.

---

> ### Author Response · Authors · 2020-11-11
> **Response to R4 - Part 1**
>
> We thank Reviewer #4 for helpful and insightful feedback. We provide answers to each question. We will also update the paper with the suggested references.
>
> **Q1.** The paper might lack some novelty. The approach seems quite incremental with respect to Yang et al. 2020 (though I am not very familiar with that work to precisely judge this). In particular, handling the additional constraint due to the baseline policy seems not that different from how the "cost constraint" is handled, and indeed the two approaches have a very similar update rule. The experimental domains seem also the same (except for the car racing one which was not present in Yang et al. 2020).
>
> **Ans:** We’d like to emphasize the main contribution of the paper: **a method to perform safe reinforcement learning when you already have a baseline/prior policy to make use of which may not satisfy safety constraints.** To our knowledge, this has not been explored in the literature and represents a novel and practically useful setting for deep RL applications (e.g., robots, drones, autonomous cars).
>
> In terms of the method, the way we handle the additional constraint due to the baseline policy is different from how the cost constraint is handled (e.g., as in Yang et al., 2020). The difference is that we need to relax the divergence constraints (for the baseline policy) due to the (likely) sub-optimality of the baseline policy. Lemma 4.1 further ensures the feasibility of the optimization problem and the exploration of the agent. Such analysis was not done in PCPO. In addition, our objective is to understand and analyze how to effectively exploit a baseline policy in constrained RL. Without such an analysis, we are not confident in deploying SPACE in real applications. Furthermore, the question of safely using baseline policies has practical potential. It is less studied by prior work.
>
> Two key contributions distinguish our work from the prior work (PCPO):
>
> **(1) Algorithmic:** The baseline policy $\pi_{B}$ need not satisfy the constraints. Hence we propose and analyze an update method for controlling the distance between the current policy and $\pi_B$ (i.e., $h_D$). Lemma 4.1 ensures the feasibility of the optimization problem and exploration of the agent (this is illustrated in Fig. 1). In addition, formulating the problem of imitating the baseline policy as a constrained optimization problem allows us to theoretically understand the effect of the baseline policy (Theorem 5.1). This was not studied in the prior work such as PCPO.
>
> **(2) Empirical:** We empirically demonstrate that the update method ensures cost satisfaction and reward improvement given the sub-optimal policies (Fig 3, Fig 4). We test SPACE on the tasks used in PCPO to ensure that we have a fair comparison and fit the context of safe RL. In addition, we demonstrate the performance of SPACE using the **human-demonstration**policy in the car-racing task. This highlights the practical use of SPACE, which was not studied in PCPO. The experiment results show that SPACE outperforms the state-of-the-art safe RL algorithm, PCPO, and the other methods that combine RL with imitation learning (e.g., d-PCPO, f-PCPO, d-CPO, f-CPO) in terms of learning efficiency.
>
> **Q2.** The setting considered for the finite-sample analysis might be overly-simplified. In particular, the analysis seems not really "finite-sample" but rather "finite-time", in the sense that it is carried out for a finite number of learning steps in a standard optimization setting where all functions (e.g., the gradients) can be exactly evaluated ...
>
> **Ans:** Thanks for pointing this out, we will revise our paper accordingly.
>
> **Q3.**  So could we simply regularize the standard RL objective with the distance wrt the baseline policy and make the regularization parameter fade over time? Would that perform worse than projecting?
>
> **Ans:** Good question! We have indeed compared SPACE with the approach that regularizes the standard RL objective with the distance with respect to the baseline policy. This approach can be found in Section 6 (d-PCPO and d-CPO). We found that  without the projection, the reward and the cost performance improves slowly. This suggests that simply adding an imitation learning objective in the reward objective is susceptible to a suboptimal $\pi_B$. In addition, formulating the problem of imitating the baseline policy as a constrained optimization problem allows us to control the distance between the baseline policy and the agent with the theoretical justification (Lemma 4.1 and Theorem 5.1).

---

> > ### Author Response · Authors · 2020-11-11
> > **Part 2**
> >
> > **Q4.** Lemma 4.1 provides a formal justification for increasing h_D over time. How is the actual increase performed in practice? Is there any fixed "increase value" or is the theoretical value from the lemma used?
> >
> > **Ans:** The value of $h_D$ depends on some nontrivial computations. Numerically, we find that this is hard to compute, so we incrementally increase the value of $h_D$ by setting a constraint, i.e., $10\times{(J_C(\pi^k)-h_C)}^2+h^k_D.$ This can be found in Appendix E.1: Implementation of Updating $h_D.$ Empirically, we found this constant does not affect the performance significantly. We will clarify this point in the updated version.
> >
> > **Q5.** Could you provide some intuition on why the finite-sample bound for SPACE with KL divergence scales with the maximum eigenvalue of F^k while the one for l2-norm scales with the minimum eigenvalue of the same matrix?
> >
> > **Ans:** Intuitively, when using KL divergence, the maximum eigenvalue of  $F^k$ determines the distance between the pre-projection and the post-projection points. On the other hand, when using L2 norm projection, the minimum eigenvalue of $F^k$ determines the distance between the pre-projection and the post-projection points. As a result, the small distance leads to a small deviation from the reward improvement direction after doing projections. This leads to more reward improvement. (This can be observed in Figure 6 in Appendix D.)
> >
> > **Q6.** It seems that the work of Yang et al. 2020 also reports a similar convergence guarantee as the one derived here. Could the authors clarify what are the technical difficulties in extending such a result to this setting? It seems to me that after the simplifications the two considered settings are quite similar.
> >
> > **Ans:** One technical challenge is to find an appropriate step size to guarantee monotonic improvement. Hence, the step size for each iteration in our paper is different from the one in PCPO (Yang et al).  In addition, in PCPO, the bound for L2 norm projection requires the assumption that the maximum eigenvalue of  $F^k$ is smaller than one. In contrast, we do not require this.
> >
> > **Q7.** In the experiments, the authors mention that "PCPO agent cannot robustly satisfy the cost constraints". Why does this happen? Doesn't PCPO handle the constraints in the same way as SPACE?
> >
> > **Ans:** This is because using the baseline policy can provide additional information about the safety constraints. For example, when using $\pi_B^{cost}$ that is only optimized for the safety constraints specified by the environment, we can observe the agent trained with SPACE satisfies the constraints more. This shows that using the baseline policy could also help to improve the cost performance (though not guaranteed). We will clarify this point in the updated version.
> >
> > **Q8.** Why is there no baseline for comparison in Figure 4?
> >
> > **Ans:** This is because in Figure 4, we focus on the effect of the sub-optimal baseline policies. Adding other baselines make the figure less readable. Addition experiments in this domain are included in Appendix E.2 (Figure 11), in which we compare SPACE with d-CPO and d-PCPO.

---

### Official Review · AnonReviewer2 · 2020-10-28
**Accelerating safe reinforcement learning with constraint-mismatched policies**

**Rating:** 6
**Confidence:** 4

**Review:**

Summary of review:
A clear problem statement & technical approach for an important problem, supported by positive empirical results, and some theoretical analysis.

Description:
The paper proposes a 3-part approach to constrained RL:   (1) perform policy optimization, (2) minize distance to the baseline policy, (3) project the policy into the policy space that satisfies the constraints.  The analysis shows better empirical performance, and a finite-sample guarantee.

Strengths:
-	The paper’s overall approach is easy to understand, and decomposes in a few conceptually simple steps.
-	I appreciated that the experiments considered tasks with both safety constraints and fairness constraints.  The latter are seldom considered in RL papers, yet very important in practice, so it’s nice to see this considered here.
-	The paper includes a good range of baseline comparisons.

Weaknesses:
-	The theoretical analysis uses several simplifying assumptions (e.g. approximation of the reward fn and constraints, and those in Assumption 1.)  What is the impact of these? Are they typically met in benchmarks?  What is the usefulness of Theorem 5.1?  I wasn’t sure what to make of this result, what it was significant or useful.  Can Eqn 12 give one intuition of when to use more / less data?
-	Key experimental details are missing.  How many seeds did you use for the experiments?  Why show standard deviation rather than standard error?  How were the hyper-parameters selected?

Questions:
-	Table 1: There are many more works on constrained MDPs. What was your inclusion/exclusion criteria for this table?
-	Why is the Disc. reward for SPACE and f-PCPO going down in Grid (Fig.3, top right plot)?  This should be explained.

---

> ### Author Response · Authors · 2020-11-11
> **Response to R2**
>
> We thank Reviewer #2 for the helpful and insightful feedback. We provide answers to individual questions below. We will also update the paper with the suggested references.
>
> **Q1.** The theoretical analysis uses several simplifying assumptions (e.g. approximation of the reward fn and constraints, and those in Assumption 1.) What is the impact of these? Are they typically met in benchmarks? What is the usefulness of Theorem 5.1? I wasn’t sure what to make of this result, what it was significant or useful. Can Eqn 12 give one intuition of when to use more / less data?
>
> **Ans:** Assumption 1 is commonly considered in the optimization literature. We agree that these assumptions may not be typically met in benchmarks (but this is also true of deep RL and the MDP assumption in general).  This is because the policy is parameterized by the neural network, which makes the constraint set non-convex. However, Theorem 5.1 is still useful since it supports the idea of starting with a small value for $h_D$ and increasing it only when needed (See Eq. 11 and Eq. 12). The smaller $H$ (the diameter of the region around $\pi_B$) is, the greater the decrease in the objective (i.e., using less data to improve the reward). In addition, it also highlights the difference between the L2 norm and KL divergence projections. This can be useful to choose between these two projections.
>
> **Q2.**  Key experimental details are missing. How many seeds did you use for the experiments? Why show standard deviation rather than standard error? How were the hyper-parameters selected?
>
> **Ans:** We used 5 random seeds for all experiments. In addition, the standard deviation is commonly used in the RL literature (e.g., PCPO, Janner et al.[1]). This allows us to make a fair comparison to the prior work. We use a grid-search to select for the hyper-parameters. We will include more details on this in the updated paper. Other experiment details can be found in the Appendix E.1.
>
> [1] When to Trust Your Model: Model-Based Policy Optimization, Janner et al., NeurIPS 2019
>
> **Q3.**  Table 1: There are many more works on constrained MDPs. What was your inclusion/exclusion criteria for this table?
>
> **Ans:** Due to the space constraints, we include the most relevant closely related algorithms. We will include more algorithms in the revised version of the paper (based on suggestions from R1).
>
> **Q4.**  Why is the Disc. reward for SPACE and f-PCPO going down in Grid (Fig.3, top right plot)?
>
> **Ans:** This is because in order to satisfy the cost constraints, the agent needs to converge to the low-reward region. Although the baselines have higher reward, they do not satisfy the constraints. In contrast, SPACE is the only algorithm that can satisfy the cost constraint in all cases.

---

### Official Review · AnonReviewer3 · 2020-10-29
**Solving CMDP faster with the help of a baseline policy**

**Rating:** 5
**Confidence:** 4

**Review:**

The paper considers the constrained Markov Decision Process (CMDP) problem where the goal is to maximize cumulative reward while satisfying the safety constraint on cumulative cost. Solving CMDP is challenging, and this paper proposes a shortcut by utilizing a given baseline policy. The idea of the proposed SPACE algorithm is to have three steps for each iteration of policy optimization. The first step is a trust region optimization step to optimize reward, the second step projects the policy to a region close to the given baseline policy, and the third step projects the policy to the constraint set.

Compared with other CMDP methods, the step of projecting close to the baseline policy allows the policy to quickly improve if the baseline policy is decent. Compared with some behavior cloning methods, the steps of reward optimization and constraint satisfaction allows the policy to achieve good final performance. I feel the ideas of the proposed SPACE algorithm is good, but I have some concerns for the current version.

- The constraints (5) and (6) in Step 2 and Step 3 need to be further discussed. From the description of Step 2, the constraint (5) should be $J_D(\pi) \leq h^k_D$ for the region around $\pi_B$. One can replace $J_D(\pi)$ by $J_D(\pi^k) + \frac{1}{\gamma} \mathbb{E}_{s\sim d^\pi, a \sim \pi}[A^{\pi^k}_D(s, a)]$ by using the performance difference formula, but the state distribution is different from the distribution in (5). It seems like (5) could be some approximation, but how it relates to the desired region and how the guarantee may be affected should be discussed. Similar for (6).

- In Lemma 4.1, only a big-O value for the update of $h^{k+1}_D$ is provided and no description on what the big-O hides. Since this value is required in SPACE, it's not clear how to implement the algorithm without the specific value. According to the proof of Lemma 4.1 in the appendix, this value seems to depend on some expectation whose computation may not be trivial.

- The practical version of SPACE described in Algorithm 1 does not solve the optimization problems (4)-(6). Since it uses (7) to update, the actual three steps of SPACE are the approximations (16)-(18) in the appendix. I think the actual steps should be stated in the main text with discussion on the effect of the approximations.

- For experiments, it is very confusing by introducing f-(P)CPO and d-(P)CPO without any explanation on the weight and the weighted objective. It is also unclear what is the value of $h_C$ and not possible to evaluate constraint violations of each methods.

- The comparisons with f-(P)CPO and d-(P)CPO is interesting, but the comparison with these baselines provides little idea for the main messages of the proposed method. SPACE is supposed to learn quicker compared to other CMDP methods by using the baseline policy, but all the compared methods actually use the baseline policy in some non-efficient way. Instead, the comparison with original CPO and PCPO will be much more meaningful as they might show the accelerated learning of SPACE, but all methods might have similar final reward/cost performance given enough training samples. Another possible comparison would be with other behavior cloning methods given a good baseline policy.


After reading the authors' response, I would like to thank the authors to clarify some of the my questions. But my main concerns still remain so I plan to keep my marginally below score. The value of the key bound in Lemma 4.1 is unknown and it's not clear about the effect of the described empirical trick. For the experiments, the proposed SPACE algorithm indeed outperforms all considered baselines. But performing better than all baselines in the final performance seem to indicate that the choice of baselines and experiment design do not present a fair comparison. One would expect the final performance of a reasonable baseline algorithm to be similar to SPACE because the idea of the proposed method is to "accelerating learning" not achieving better final performance. The poor final performance of the baselines seems to be due to the inappropriate adaption of existing methods.

---

> ### Author Response · Authors · 2020-11-11
> **Response to R3**
>
> We thank Reviewer #3 for helpful and insightful feedback. We provide answers to each question. We will also update the paper with the suggested references.
>
> **Q1.** The constraints (5) and (6) in Step 2 and Step 3 need to be further discussed…
>
> **Ans:** Yes, the state distribution is different from the distribution in (5). The reason why we can use this approximation is that for each update, we only use a small step size $\delta:$ $E_{s\sim d^{\pi^k}}[D_{KL}[\pi(s)\|\|\pi^k(s)]]\leq\delta.$ This approximation is also used in CPO (Achiam et al., ICML2017). In addition, we empirically observe that this approximation does not affect the performance. Furthermore, the guarantee in Lemma 4.1 is based on the distribution $d^{\pi^k}$ and *has considered the mismatch of the state distributions by incorporating the KL divergence term between two policies* (See Appendix B, the KL term in the first inequality). So Lemma 4.1 still holds.
>
> **Q2.** Discussion on Big O in Lemma 4.1.
>
> **Ans:** Yes, you are right in that the exact value of $h_D$ depends on some nontrivial computations. Numerically, we find that this is hard to compute, so we incrementally increase the value of $h_D$ by setting a constraint, i.e., $10\times{(J_C(\pi^k)-h_C)}^2+h^k_D.$ This detail can be found in Appendix E.1: Implementation of Updating $h_D$ - we will add a pointer to this in the main paper. Empirically, we found this constant does not affect the performance significantly.
>
> **Q3.** The practical version of SPACE described in Algo. 1 does not solve the optimization problems (4)-(6)...
>
> **Ans:** Thank you for pointing this out. We will include the actual steps and the discussion about the approximation in the main text.
>
> **Q4.** For experiments, it is very confusing by introducing f-(P)CPO and d-(P)CPO without any explanation on the weight and the weighted objective. It is also unclear what is the value of $h_C$ and not possible to evaluate constraint violations of each method.
>
> **Ans:** f-(P)CPO and d-(P)CPO add the weighted divergence objective (imitating the baseline policy) in the reward function. These baselines simply regularize the standard RL objective with the distance with respect to the baseline policy and make the weighted parameter smaller over time. These approaches are commonly used in the context of imitation learning (e.g., Rajeswaran et al.). For all the experiments and the algorithms, the weight is fixed and it is set to 1. We will include this in the draft.
>
> The value of $h_C$ for each task is included in Appendix E.1, table 2. In addition, in Figure 3, the black dashed line is the value of $h_C.$ We will make this clear in the main text.
>
> **Q5.** The comparisons with f-(P)CPO and d-(P)CPO is interesting, but the comparison with these baselines provides little idea for the main messages of the proposed method. SPACE is supposed to learn quicker compared to other CMDP methods by using the baseline policy, but all the compared methods actually use the baseline policy in some non-efficient way. Instead, the comparison with the original CPO and PCPO will be much more meaningful as they might show the accelerated learning of SPACE, but all methods might have similar final reward/cost performance given enough training samples. Another possible comparison would be with other behavior cloning methods given a good baseline policy.
>
> **Ans:**  f-(P)CPO and d-(P)CPO are variants of the algorithm used in prior work (Rajeswaran et al.). This comparison highlights the effectiveness of SPACE and the disadvantages of these commonly-used approaches. In addition, since SPACE and PCPO optimize the same reward function and the distance between the agent and the baseline policy increases during training, they converge to the same degree of the reward.
>
> We have compared SPACE with the **state-of-the-art**safe RL algorithm, PCPO in Fig. 3. We find that SPACE has better training efficiency than the original PCPO.
>
> Behavior cloning methods do not consider safety. In addition, our setting is to learn from the potential **sub-optimal**baseline policies (as stated in the title). As a result, we emphasize that behavior cloning methods given a good $\pi_B$ are not the main focus of our setting.
>
> However, we have indeed included a variant of behavior cloning methods in Appendix E.2. (Fixed $h_D$, Fig. 10). This approach is similar to SPACE, but with the fixed $h_D$ (i.e., the agent stays close to the baseline policy throughout training). This fixed $h_D$ is similar to behavior cloning since it imitates the baseline policy at every step. We observe that SPACE outperforms this behavior cloning method in terms of reward performance given the baseline policy that optimized for the reward. This is because the agent trained with this behavior cloning method is prone to violating the constraints, leading to finding a suboptimal policy.
>
> Rajeswaran et al. Learning complex dexterous manipulation with deep reinforcement learning and demonstrations. RSS, 2017.

---

### Official Review · AnonReviewer1 · 2020-10-29
**Good paper, some aspects to discuss and improve.**

**Rating:** 7
**Confidence:** 4

**Review:**

update after rebuttal: the authors answered all my questions to my satisfaction.

* Summary and Contribution

This work presents a novel approach to safe reinforcement learning. In particular, in line with several other works, the others propose to start from a baseline policy, which is then improved. Three key steps are identified: (1) updating the baseline to maximize expected reward, (2) measuring and controlling the ``distance'' between the new policy and the baseline, and (3) projecting the policy to satisfy given safety constraints. The authors provide both the an theoretical analysis and a number of experiments.


Reasons for Score
I believe this is a nice paper with good contribution and valid evaluation, but some parts can be improved.

Strengths

- A novel method for safe RL
- The baseline policy is not required to already satisfy the safety constraints
- A thorough theoretical analysis
- Convincing, though small-scale, experiments

Weaknesses

- Some related work should be discussed. The statement that all other approaches using a baseline do require a safe baseline is not entirely true and needs further detailed discussion.
- Better motivation why this is Safe RL -> How to 'safely' obtain a baseline?
- The approach seems like a (nice) collection of already established building blocks.


Questions for Authors

- I would like a better discussion on why it is realistic to have a baseline policy. In a (physical) safety-critical scenario, any baseline may be either so bad that the approach is not effective anymore, or it may be obtained using actual unsafe interaction with the environment.

- A key step of the approach seems to be 'Trust region policy optimization' work, which is well known. How would you argue that your work is really novel, and not just reusing that approach?

- The paper states that other approaches using a baseline require it to be safe. I do not think that is true, see the detailed comment.

- Please compare your work to the literature on model repair, see the detailed comments.


Detailed comments

- safe baselines, please discuss:

[1] makes two assumptions: a set of safe states is given, and there is regularity for the safety function. My understanding is that these assumptions allow the agent to explore within the safe region until it learns enough to explore new areas. Also, [2] and [3] seem to make the same assumptions. [3] states that "Our approach [...] does not require a backup policy but only a set of initially safe states from which the agent starts to explore."



[1] Akifumi Wachi, Yanan Sui 2020
   Safe Reinforcement Learning in Constrained Markov Decision Processes.
   In International Conference on Machine Learning (ICML) 2020

[2] Sui, Y., Gotovos, A., Burdick, J. W., and Krause, A.
  Safe exploration for optimization with Gaussian processes.
  In International Conference on Machine Learning (ICML), 2015

[3] Turchetta, M., Berkenkamp, F., and Krause, A.
  Safe exploration in finite Markov decision processes with Gaussian processes.
  In Neural Information Processing Systems (NeurIPS), 2016.


- model repair, please discuss

In the essence, and admittedly, the work [4,5]listed below is from the formal methods community, so it may just not be known for ICLR. Yet, what happens: A model (for instance, a policy), is 'repaired' to satisfy (temporal logic) safety constraints, with minimal distance to the original policy. At least a discussion is in order.

[4] Shashank Pathak, Erika Ábrahám, Nils Jansen, Armando Tacchella, Joost-Pieter Katoen:
A Greedy Approach for the Efficient Repair of Stochastic Models. NFM 2015: 295-309

[5] Ezio Bartocci, Radu Grosu, Panagiotis Katsaros, C. R. Ramakrishnan, Scott A. Smolka:
Model Repair for Probabilistic Systems. TACAS 2011: 326-340


- general related work:

There is some other related work both on Safe RL that could be discussed.


Prashanth L, A., and Michael Fu. "Risk-sensitive reinforcement learning: A constrained optimization viewpoint." arXiv preprint arXiv:1810.09126 (2018)

Zheng, Liyuan, and Lillian J. Ratliff. "Constrained upper confidence reinforcement learning." arXiv preprint arXiv:2001.09377 (2020).

Sebastian Junges, Nils Jansen, Christian Dehnert, Ufuk Topcu, Joost-Pieter Katoen:
Safety-Constrained Reinforcement Learning for MDPs. TACAS 2016: 130-146

Nils Jansen, Bettina Könighofer, Sebastian Junges, Alex Serban, Roderick Bloem:
Safe Reinforcement Learning Using Probabilistic Shields. CONCUR 2020: 3:1-3:16

Guy Avni, Roderick Bloem, Krishnendu Chatterjee, Thomas A. Henzinger, Bettina Könighofer, Stefan Pranger: Run-Time Optimization for Learned Controllers Through Quantitative Games. CAV (1) 2019: 630-649

Mohammadhosein Hasanbeig, Alessandro Abate, Daniel Kroening:
Cautious Reinforcement Learning with Logical Constraints. AAMAS 2020: 483-491

---

> ### Author Response · Authors · 2020-11-11
> **Response to R1 - Part 1**
>
> We thank Reviewer #1 for the helpful and insightful feedback. We provide answers to individual questions below. We will also update the paper with the suggested references.
>
> Response to Weaknesses:
>
> **W1.** The statement that all other approaches using a baseline do require a safe baseline is not entirely true and needs further detailed discussion.
>
> **Ans:** Thanks for pointing this out. Please see our discussions Q3 and Q4.
>
> **W2.** Better motivation why this is Safe RL -> How to 'safely' obtain a baseline?
>
> **Ans:** In many real-world applications such as drones, we can obtain baseline policies modeled from the first principle physics, or pretrain baseline policies in the constrained and safe environment, or use rule-based baseline policies. Importantly, like you mention in your other comments, we _do not_ assume the baseline has to be a ‘safe policy’ - it can be a heuristic that ignores safety constraints. This is one of the main motivations for our algorithm: to utilize priors from the baseline which may be unsafe, but guarantee the safety of the newly learned algorithm according to the provided constraints. We will update the paper with more discussion to clarify this.
>
> **W3.** The approach seems a (nice) collection of already established building blocks.
>
> **Ans:** SPACE builds upon PCPO and TRPO. However, our objective is to understand and analyze how to effectively exploit a baseline policy in constrained RL. This is an important step in deploying reinforcement learning algorithms in real-world applications. Two key contributions distinguish our work from the prior work.
> (1) Algorithmic: The baseline policy $\pi_{B}$ need not satisfy the constraints. Hence we propose and analyze an update method for controlling the distance between the current policy and $\pi_B$ (i.e., $h_D$). In addition, formulating the problem of imitating the baseline policy as a constrained optimization problem allows us to theoretically understand the effect of the baseline policy (Theorem 5.1). This was not studied in the prior work such as PCPO or TPRO.
> (2) Empirical: We empirically demonstrate that the update method ensures cost satisfaction and reward improvement. We compared SPACE with and without using the update method of $h_D$ in Appendix E.2 (Fixed $h_D$ (see Fig. 10)). We observe that SPACE yields a greater reward given a suboptimal $\pi_B.$ In contrast, the direct use of $\pi_B$ yields more cost violations. We further demonstrate the performance of SPACE using the human-demonstration policy in the car-racing task. This highlights the practical use of SPACE, which was not demonstrated in PCPO.

---

> > ### Author Response · Authors · 2020-11-11
> > **Part 2**
> >
> > Response to Questions:
> >
> > **Q1.** I would like a better discussion on why it is realistic to have a baseline policy. In a (physical) safety-critical scenario, any baseline may be either so bad that the approach is not effective anymore, or it may be obtained using actual unsafe interaction with the environment.
> >
> > **Ans:** This is a good point, and we agree that the baseline policy could be very bad. However, due to our method of using projections and relaxation of the imitation constraints in the policy space, it cannot hurt the learning process and often provides a nice boost to learning. In the worst case, the baseline does not provide any advantage over learning without one, but more often than not, we can always write down some heuristics/rules that would help as a baseline.
> >
> > For example, when we want to learn a policy for controlling a drone, we can obtain a baseline policy from the first principle physics given the reward objective and safety constraints. This is often done in the control community (Fisac et al., 2018, Shi et al., 2019). Due to a change in the environment, the baseline policy may not be efficient or safe enough in some situations such as windy environments where one does not want the drone to crash into objects. In such a case, we can use SPACE to refine the policy.
> >
> > We agree that in some safety-critical scenarios, it may be hard or even impossible to obtain the baseline policy safely. However, we can try to design a simulator by observing the system dynamics and then obtain baseline policies trained in the simulator. An exploration of different ways of obtaining useful baseline policies and their effectiveness are great suggestions for future work.
> >
> > Fisac et al., A general safety framework for learning-based control in uncertain robotic systems, 2018
> > Shi et al., Neural Lander: Stable Drone Landing Control using Learned Dynamics, 2019
> >
> > **Q2.** A key step of the approach seems to be 'Trust region policy optimization' work, which is well known. How would you argue that your work is really novel, and not just reusing that approach?
> >
> > **Ans:** Our method is novel because it uses two projection steps in addition to constrained policy improvement like TRPO - 1) projecting w.r.t the baseline policy and 2) projecting w.r.t the safety constraints. Our novelty lies in providing an alternating optimization scheme that effectively uses all the different signals available to the agent during learning. Please also see our response to Q3.
> >
> > **Q3.** Discussion on the use of safe baseline policies in the related work section.
> >
> > **Ans:** Thanks for the related work mentioned here. We will update the related work section and cite these papers. As you point out, the assumption of these works [1][2][3] is that the initial safe set is given, and the agent explores the environment and verifies the safety function from this initial safe set. There is no baseline policy here. In contrast, our assumption is to give a baseline policy to the agent. Both assumptions are reasonable as they provide an initial understanding of the environment.
> >
> > **Q4.** Discussion on the model repair.
> >
> > **Ans:** Thanks for pointing this out. While our work shares the same concept as [4][5] (modifying the existing known solutions based on new conditions), we solve this problem using projections in the policy space. We will update the related work section and cite these papers.
> >
> > **Q5.** There is some other related work both on Safe RL that could be discussed.
> >
> > **Ans:** Thank you for pointing this out, we will update the related work section with a discussion on these.

---

### Author Response · Authors · 2020-11-11
**General Response**

We thank all the reviewers for their thorough and very helpful feedback. We are currently working on revising the paper and will upload an updated version soon. In the meantime, we have provided detailed responses to all your questions below - please take a look and let us know if you have any further questions/comments!

---

> ### Author Response · Authors · 2020-11-11
> **Updated Version**
>
> We have uploaded an updated version based on the comments. The modified part is shown in red. Please let us know if you have any further questions/comments!

---

### Decision · Program_Chairs · 2021-01-07
**Final Decision**

**Decision:**

Reject

**Comment:**

The paper is about a reinforcement learning algorithm that operates in a Constrained MDP and is provided with a baseline policy.
Although the reviewers acknowledge that the paper has some merits (well-written, clearly organized, significant empirical evaluation, reproducible experimental results), some concerns have been raised about the novelty of the proposed solution and of its theoretical analysis. The reviewers feel that the authors' responses have not properly addressed all their doubts.
The paper is borderline and I think that it is not ready for publication in the current form.
I encourage the authors to update their paper following the reviewers' suggestions and try to submit it in one of the forthcoming machine learning conferences.